**Subject Category:**
Biology (whole organism)

taxonomy and systematics

conservation management, extinction, temperate grasslands, taxonomy, threatened species, *Tympanocryptis pinguicolla*

**Author for correspondence:**
Jane Melville
e-mail: jmelv@museum.vic.gov.au

# Taxonomy and conservation of grassland earless dragons: new species and an assessment of the first possible extinction of a reptile on mainland Australia

Jane Melville[1,2,3], Kirilee Chaplin[1,2], Mark Hutchinson[4], Joanna Sumner[1], Bernd Gruber[3], Anna J. MacDonald[3] and Stephen D. Sarre[3]

[1]Department of Sciences, Museums Victoria, Carlton Gardens, Victoria 3052, Australia
[2]School of Biosciences, University of Melbourne, Parkville, Victoria 3052, Australia
[3]Institute for Applied Ecology, University of Canberra, Canberra, Australian Capital Territory 2601, Australia
[4]South Australian Museum, North Terrace, Adelaide, South Australia 5000, Australia

JM, 0000-0002-9994-6423

Taxonomic research is of fundamental importance in conservation management of threatened species, providing an understanding of species diversity on which management plans are based. The grassland earless dragon lizards (Agamidae: *Tympanocryptis*) of southeastern Australia have long been of conservation concern but there have been ongoing taxonomic uncertainties. We provide a comprehensive taxonomic review of this group, integrating multiple lines of evidence, including phylogeography (mtDNA), phylogenomics (SNPs), external morphology and micro X-ray CT scans. Based on these data we assign the lectotype of *T. lineata* to the Canberra region, restrict the distribution of *T. pinguicolla* to Victoria and name two new species: *T. osbornei* sp. nov. (Cooma) and *T. mccartneyi* sp. nov. (Bathurst). Our results have significant conservation implications. Of particular concern is *T. pinguicolla*, with the last confident sighting in 1969, raising the possibility of the first extinction of a reptile on mainland Australia. However, our results are equivocal as to whether *T. pinguicolla* is extant or extinct, emphasizing the immediate imperative for continued surveys to locate any remaining populations of *T. pinguicolla*. We also highlight the need for a full revision of conservation management plans for all the grassland earless dragons.

# 1. Introduction

The recent proposal to establish a formalized system for the oversight of taxonomic change [1], has generated heated debate over the role and governance of taxonomy for biodiversity conservation. While the proposal has been widely criticized [2], the message that rigorous scientific-based taxonomic research is an essential element in the conservation of species has emerged from the subsequent debate. Such taxonomic research is fundamental to conservation managers and researchers alike, particularly in defining groups that harbour taxonomic uncertainty. The underestimation of species diversity across numerous vertebrate groups is substantial, with a sizeable proportion of species comprising undiagnosed complexes that hide taxa [3] and are therefore invisible to the IUCN Red List and other conventional conservation assessments. The two main causes of undiagnosed complexes is the presence of cryptic species, consisting of phenotypically indistinguishable taxa [4], or simply a lack of rigorous taxonomic research in the target group [5]. The critical importance of alpha taxonomy in such groups to prevent further biodiversity losses cannot be overemphasized [6].

Globally, 31 reptiles have been listed as extinct or extinct in the wild on the IUCN Red List but until now there have been no recorded reptile extinctions on the Australian mainland. One Australian reptile clade, the agamid genus *Tympanocryptis*, is noteworthy in this regard as it includes several species with very restricted or fragmented distributions [7]. Studies of the *T. tetraporophora* and *T. cephalus* species complexes [7–9] clarified species boundaries in these two clades, showing that *Tympanocryptis* provides examples of both neglected taxonomy and the presence of cryptic species, and in doing so revealed species in which some, or all, populations are regarded as under severe threat of extinction. The work completed on the *T. tetraporophora* and *T. cephalus* complexes leaves a paraphyletic set of named but not always well-characterized taxa that have unexplained variation, uncertain specific distinctness and, in some cases, strong conservation concerns. Our study now addresses the southeastern component to that complex that includes the type species of *Tympanocryptis*, *T. lineata*, plus *T. pinguicolla*, which first entered the literature as a subspecies of *T. lineata*. This group had been of particular conservation concern, occurring in fragments of the temperate grasslands of southeastern New South Wales, Australian Capital Territory and Victoria [10–15].

As a foundation to this work we provide a comprehensive phylogeographic study, based on approximately 1800 bp mtDNA, of all lizards that have been assigned to the '*T. lineata*' complex, including former and current subspecies: *T. lineata*, *T. pinguicolla*, *T. houstoni*, *T. centralis* and *T. lineata macra*. We then use this phylogeographic framework to assess lizards assigned to the southeastern Australian grassland taxa in a whole-evidence taxonomic revision, incorporating genomics, external morphology, geometric morphometric analysis of micro X-ray CT scans and a review of historical records. Our study reveals that this group, formerly regarded as four disjunct populations of *T. pinguicolla* [13], comprises four distinct species that are confined to small geographical areas. We also revisit the nomenclatural details of these lizards, and show that the oldest available name for these lizards is *T. lineata*, not *T. pinguicolla*. Finally, after providing a taxonomic treatment of this group, which we refer to as the grassland earless dragons (GEDs), we undertake an assessment of the likelihood of *T. pinguicolla* being extinct versus extant based on sighting data and records, and discuss the conservation implications of our study.

Our approach highlights the power of rigorous taxonomic research and its central role in conservation. The first step in a comprehensive taxonomic assessment such as ours, is determining to which lineage the available names belong. This can often be difficult where the locality data recorded in the historic species descriptions is non-specific, as is the case with *T. lineata*.

## 1.1. *Tympanocryptis lineata* lectotype

As part of this study we have had to consider the identity of the type species and oldest specific name assignable to the genus, *Tympanocryptis lineata* Peters, 1863. The result of our application of the rules of nomenclature (a formality in most cases) has been an unanticipated but unavoidable transfer of the name from the population with which all literature has associated this binomial since its original description [16,17] to one of the grassland earless dragons, which are the main focus of this paper. Given that this change was unforeseen by those who work with or are familiar with these animals, we outline the documentary and historical evidence that made this change inevitable in electronic supplementary material, appendix S1. Our morphological examination of the specimen accords completely with the historical data, confirming that the lectotype specimen derives from one of the southeastern Australian grassland earless dragons (detailed in the Results section). To further determine the precise

R. Soc. open sci. **6**: 190233

locality from which ZMB 740 was collected (the specimen lacks collection data) and thus determine the lineage to which the name *T. lineata* should be assigned, we include this specimen in our multivariate assessment of external morphology below.

# 2. Methods

## 2.1. Specimen information

A list of all specimens used in this study is in electronic supplementary material, table S2. Specimens from Museums Victoria, South Australian Museum, Australian National Wildlife Collection and Australian Museum were used in genetic, external morphology and osteological geometric morphometric analyses. Where available, genetic data from previously sequenced specimens were used, with GenBank numbers provided. We also included type material for the available names in order to stabilize the nomenclature of the group.

The electronic edition of this article conforms to the requirements of the amended International Code of Zoological Nomenclature (ICZN), and hence the new names contained herein are available under that Code from the electronic edition of this article. This published work and the nomenclatural acts it contains have been registered in ZooBank. The LSID for this publication is: urn:lsid:zoobank. org:pub:7FF07B9E-956D-4D0E-B3B2-24E989D91720.

## 2.2. Phylogeography

### 2.2.1. DNA sequencing and alignment

Genomic DNA was isolated from liver samples using Proteinase K digestion and chloroform-isoamyl alcohol extraction or using a DNAeasy tissue extraction kit (Qiagen) as per manufacturer's instructions. For all specimens, a fragment (approx. 1800 bp) of the mtDNA genome was targeted that includes the entire protein-coding gene *ND2* (NADH dehydrogenase subunit two) and flanking genes encoding tRNA$^{Trp}$, tRNA$^{Ala}$, tRNA$^{Asn}$, tRNA$^{Cys}$, tRNA$^{Tyr}$. Oligonucleotide primer pairs used in PCR amplification and sequencing of the mtDNA target region are detailed in Melville *et al.* [7]. Amplifications were performed in 25 µl volumes in the presence of 1.5 mM MgCl$_2$, 0.2 mM dNTPs, 0.2 µM of forward and reverse primer, 1× Qiagen PCR buffer and 1 unit of HotStarTaq DNA polymerase (Qiagen). Thermal cycling conditions consisted of an initial denaturing and enzyme activation step at 95°C for 15 min followed by 40 cycles of denaturing at 95°C for 20 s, annealing at 55°C for 20 s, and extension at 72°C for 90 s, using a Corbett thermocycler. Negative controls were run for all amplifications. PCR amplifications were visualized on a 1.2% agarose mini-gel or using a QIAxcel gel analysis system. Amplified products were purified using GFX spin columns or using SureClean Plus (BIOLINE). Purified product was sent to Macrogen (Korea) for sequencing.

Sequence chromatograms were edited using Geneious 6.1.2 (Biomatters Ltd) to produce a single continuous sequence for each specimen. These sequences and previous GenBank sequences were aligned and protein-coding regions were translated to amino acids to check alignment and for stop codons.

### 2.2.2. Phylogenetic analysis

The ND2 coding region and flanking tRNA regions were found to follow the GTR+I+G model of substitution with no partitioning schemes using the corrected Akaike information criterion (AICc) on PartitionFinder2 on the CIPRES Science Gateway [18,19]. Bayesian analysis was performed using MrBayes [20] on the CIPRES Science Gateway, with two runs of four independent MCMC chains (each 50 000 000 generations long, sampled every 1000 generations), under a GTR+I+G model with flat priors. Tracer v. 1.6 [21] was used to check for stationarity and convergence of the chain outputs. The trees were subject to a 25% burn-in in MrBayes, summarized and posterior probabilities obtained. Pairwise mean uncorrected genetic distance between described and putative species was calculated in Mega 7 [22].

An additional phylogenetic analysis, using methods described above, was conducted on a subsection of the dataset, using 11 grassland earless dragon samples and *T. houstoni* as an outgroup. In this analysis we only included the 159 bp segment of the ND2 protein coding gene that had been successfully sequenced in species B from a frozen specimen. This analysis was used to confirm that phylogenetic relationships could be ascertained from this small region of mtDNA. We also calculated uncorrected pairwise genetic distance

between lineages using this small fragment of DNA to ensure it was similar to that for the approximately 1800 bp region of mtDNA used in the full analysis.

## 2.3. Phylogenomics

Genomic data were obtained for 96 specimens from Cooma ($N = 27$), northern Canberra ($N = 24$) and southern Canberra ($N = 44$) using 15 µl volumes of high concentration DNA samples. These were sent to Diversity Arrays Technology (Canberra, Australia) for digestion using genus-specific restriction enzymes, ligation, amplification and sequencing on an Illumina HiSeq2500 (California, USA), in a genome complexity reduction method optimized for *Tympanocryptis* species and which has previously been used to distinguish species of Australian Agamidae [23]. The genome sequence of the related *Pogona vitticeps* was used as the reference alignment [24]. The preliminary pipeline applied filters to ensure quality and reproducibility of markers, and a secondary pipeline using DArTseq proprietary calling algorithms scored the genotypes of polymorphic loci across all individuals. The loci were combined into a matrix of SNP genotypes for individuals, with integers representing genotype states compared to the reference; homozygous; heterozygous and alternate homozygous.

SNP data were imported into R v. 3.3.3 [25] and population locality metadata assigned with the R package 'dartR' [26]. The 'dartR' package was used to filter the data for all subsequent analyses under several parameters; callrate by individual > 0.90, callrate by locus > 0.95, monomorphic loci = 0, reproducibility = 1, and loci out of Hardy–Weinberg equilibrium or under linkage disequilibrium ($F_{ST} > 0.9$). Principle coordinates analysis (PCA) was run to visualize the separation of each population, and a Mantel test was used to assess isolation by distance. We performed pairwise $F_{ST}$ (by population) analysis of the filtered data with 100 bootstraps using 'StAMPP' [27], and calculated $G'_{ST}$ and Jost's D, using 'mmod' [28].

## 2.4. External morphology

Seventeen meristic and metric characters previously used in *Tympanocryptis* taxonomy [7,9] and thought to be potentially diagnostic were recorded for all GED species. Vouchers were putatively assigned to species based on mtDNA (where available), geographical location and appearance. Electronic callipers were used for all morphological measures to the nearest 0.1 mm and all bilateral counts and measurements were recorded on the left side (where possible). Morphological measurement of *T. lineata* Lectotype ZMB 740 were taken from high-resolution photographs that included a scale, using software NIH ImageJ [29]. All external morphology analyses were run in SYSTAT v. 13.2. Non-mature individuals and specimens with missing data were removed from these analyses. A linear regression was performed on snout–vent length against all other measurements to standardize for size, and the residuals used in further analyses. An initial discriminant functional analysis (DFA) was conducted on all GED voucher specimens, to determine which lineage the *T. lineata* Lectotype ZMB 740 was most similar to morphologically, based on: tail length, rear leg length, front leg length, snout width, head width and neck width. All vouchers were included in this initial DFA to provide the full range of morphological variation (both male and female). Mahalanobis distances from group means were calculated and a posterior probabilities test for group membership was used to assess the probability of ZMB 740 belonging to each of the four geographical groups. Subsequently, data were split into males and females and analysed separately, with DFAs performed for each sex for all study species.

## 2.5. Osteological geometric morphometrics

The cranium of specimens were scanned using a 180 kV nanofocus tube in a Phoenix Nanotom M machine (GE Measurement & Control, Massachusetts, USA) with a tungsten target, for 600 projections at 55 kV and 400 µA for 500 ms. The final voxel size was 15 µm. Volumetric reconstructions of the skulls were generated by datos|x-reconstruction software (GE Sensing & Inspection Technologies GmbH, Wunstorf, Germany) and three-dimensional surface models were subsequently prepared in VGStudio Max2.1 (Volume Graphics, Heidelberg, Germany). Forty-nine landmarks were chosen to form a holistic representation of overall cranial shape, and were placed ventrally, dorsally, laterally and posteriorly across the surface models using Landmark Editor v. 3.6 (Institute for Data Analysis and Visualization, UC Davis, USA). Three-dimensional landmark coordinates were exported to MorphoJ [30] and subjected to a generalized Procrustes fit to align and orient the points and standardize for size relative to their centroid.

The generalized Procrustes coordinates were used for subsequent data analysis, using the R v. 3.3.3 package 'geomorph' [31] unless otherwise stated, where statistical tests of significance were subject to 10 000 iterations under a randomized residual permutation scheme. ANOVAs testing for effects of sex or centroid size were conducted to ensure that there was no results bias caused by sexual dimorphism or specimen size. An ANOVA was used to test for a significant interaction among all species, followed by pairwise tests to identify differences between each species pair. A preliminary PCA was conducted in MorphoJ, and the PC1 and PC2 scores for each landmark were assessed to determine which characters contributed the largest amount of morphological variation. The negative and positive extremes of the variation of each PC axis were used to warp a general mesh to graphically represent the morphological variation observed and explained by PC1 and PC2. A subsequent ANOVA was used to test for a significant effect of species, followed by pairwise tests to identify differences among populations.

## 2.6. Estimating chance of extinction for Tympanocryptis pinguicolla

Following our taxonomic revision restricting the distribution of T. pinguicolla to Victoria, we estimated the chance that it is now extinct using sighting and museum records as a time series (see electronic supplementary material, table S3). Dates for museum records were confirmed by checking original hand-written registers held at Museums Victoria. We used five estimators that vary in their levels of complexity [32] and were calculated using the R package 'sExtinct' [33]. We applied these analyses to two datasets: (i) a dataset consisting of confirmed sightings only based on museum voucher specimens and sightings with high confidence based on advice from researchers conducting survey work on T. pinguicolla; and (ii) a dataset including all records from the Victorian Biodiversity Atlas (VBA), which include more recent unconfirmed sightings for which there is a lower level of confidence (these are discussed further in the Taxonomic revision section below). We used five methods to estimate possible extinction dates. These included an approach focused on the discrepancy between the observed interval of sightings (between the first and last sighting) [34], a second method developed by Solow [35] that assumes sightings are a stationary Poisson process and a subsequent formulation that makes the more accurate assumption that sightings actually follow a truncated exponential distribution, declining until extinction [36]. Finally, we used an optimal linear estimator (OLE) method, which is the most widely used and robust non-parametric extinction estimator [37]. We use an alpha cut-off of $\alpha < 0.05$, which is equivalent to a 95% chance of extinction, by the year 2019; the OLE estimator provides the year at which extinction occurs rather than an estimate of '$\alpha$'.

# 3. Results

## 3.1. Morphological placement of Tympanocryptis lineata Lectotype ZMB 740

Visual inspection of Tympanocryptis lineata Lectotype ZMB 740 confirms that, based on morphology, it is a member of the grassland earless dragons (GEDs) of southeastern Australia. The specimen, an adult female, has a prominent five-lined pattern that crosses six dark dorsal crossbands, possesses long very spinous enlarged dorsal scales and a lateral skin fold, and lacks enlarged tubercular scales on the thighs, a combination that restricts its origin to either the Canberra area or the Monaro tableland (both visited by Lhotsky on his explorations [38]). We then undertook a discriminant functional analysis (DFA) including all GED voucher specimens of known locality to determine which lineage the T. lineata Lectotype ZMB 740 belonged to morphologically, based on: tail length, rear leg length, front leg length, snout width, head width and neck width.

The DFA of the four grassland earless dragons significantly distinguished geographical groups (Wilks' $\lambda_{6,3,28} = 0.052$, $F_{18,65} = 6.67$, $p < 0.001$), grouped as Melbourne, Cooma, Bathurst and Canberra. The DFA correctly classified 78% of animals into the assumed a priori geographical groups (figure 1). Canonical factor 1 explained 75.8% of variance and canonical factor 2 explained 22.7%. When corrected for within-group variance, both canonical factors were positively associated with tail length and snout width, and canonical factor 1 was also negatively associated with hindlimb length (figure 1). Thus, animals from the Bathurst region have longer tails, wider snouts and shorter hindlimbs, while the Canberra animals have longer tails, wider snouts and somewhat longer hindlimbs. Mahalanobis distance from group means with a posterior probabilities test for group

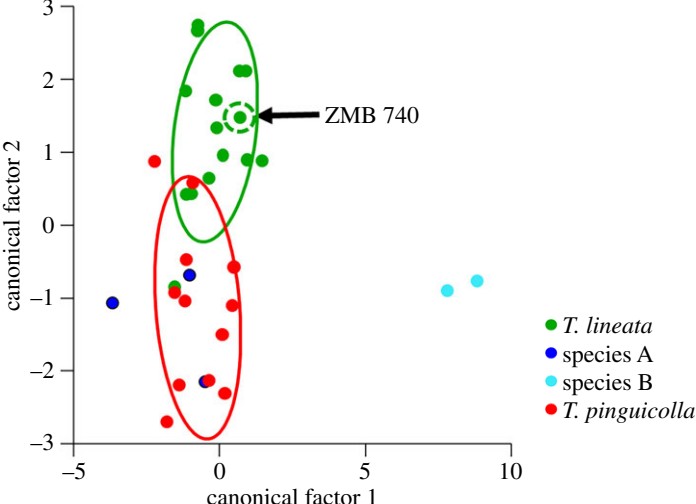

**Figure 1.** Discriminant functional analyses of grassland earless dragons, using six external morphology measurements. The position of the *Tympanocryptis lineata* Lectotype ZMB 740 is indicated with an arrow.

membership was used to assess the probability of ZMB 740 belonging to each of the four geographical groups. ZMB 740 was significantly unlikely to belong to the Melbourne (Mahalanobis squared distance = 18.78, $p = 0.013$), Cooma (Mahalanobis squared distance = 21.86, $p = 0.003$) or Bathurst (Mahalanobis squared distance = 72.30, $p < 0.001$) groups but could not be rejected as belonging to the group of voucher specimens from the Canberra region (Mahalanobis squared distance = 10.05, $p = 0.985$). Thus, based on external morphological measurements it is most likely that ZMB 740 originates from the Canberra populations of GEDs. This means that the name *Tympanocryptis lineata*, with Lectotype ZMB 740 being the primary type specimen, applies to the Canberra populations of GEDs. Based on this, we refer to the Canberra populations from here on in as *T. lineata* and the other three putative species of GEDs as *T. pinguicolla* (Melbourne), species A (Cooma) and species B (Bathurst).

## 3.2. Species delimitation

### 3.2.1. Phylogeography

To investigate phylogenetic relationships among the *Tympanocryptis lineata* group, including all previous and current subspecies and junior synonyms, we used Bayesian inference to undertake a phylogenetic analysis of mtDNA (ND2) sequence obtained from 290 *Tympanocryptis* samples as well as samples from two other Australian agamid genera (*Pogona* and *Rankinia*) as outgroups (figure 2). The alignment comprised 1840 characters with 862 being parsimony informative. A GTR + I + Γ model was selected as the best fitting model using AIC. The analysis presented here also includes sequence (160 bp) of two historical specimens: (i) a frozen earless dragon collected in Bathurst, NSW, during the 1990s (species B); and (ii) a previously published sequence of a historical museum specimen of *T. pinguicolla* [40].

The mtDNA Bayesian phylogeny (mean ln-likelihood −25702.85) strongly supported the genus *Tympanocryptis* as a monophyletic lineage (posterior probability 100%). Within *Tympanocryptis* the samples comprising *T. lineata, T. houstoni, T. pinguicolla, T. l. centralis* and *T. l. macra*, all of which first entered the literature as subspecies of *T. lineata*, were paraphyletic. These named taxa fell into three major phylogenetic lineages within *Tympanocryptis* (figure 2a): (i) *T. l. macra*, from northern Australia, forming a basal clade within *Tympanocryptis* (posterior probability 100%); (ii) *T. lineata* (Canberra), *T. houstoni, T. pinguicolla*, belonging to a diverse eastern/southern Australian lineage (posterior probability 100%); and (iii) *T. centralis*, belonging to the pebble dragon clade, which is widely distributed in stony deserts across the arid zone (posterior probability 100%).

Within the eastern/southern Australian lineage (figure 2a), which currently includes *T. lineata* (Canberra), *T. houstoni* and *T. pinguicolla*, phylogenetic analyses resolved nine highly supported clades (posterior probability greater than 98%). This includes two historic samples for which only single

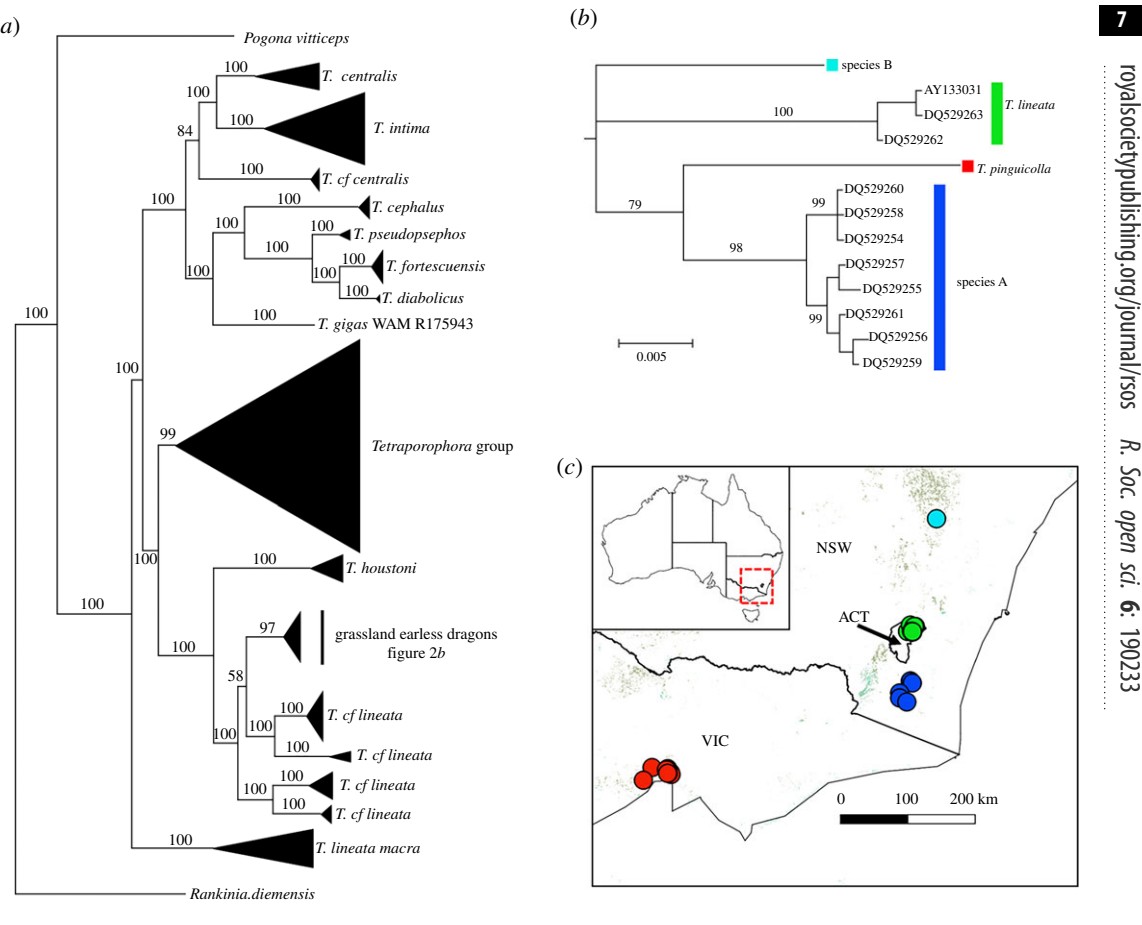

**Figure 2.** Bayesian phylogeny of described and putative *Tympanocryptis* species using approximately 1800 bp of the mtDNA sequence including the gene *ND2* (NADH dehydrogenase subunit two) and flanking genes encoding tRNA$^{Trp}$, tRNA$^{Ala}$, tRNA$^{Asn}$, tRNA$^{Cys}$, tRNA$^{Tyr}$. Posterior probabilities are indicated on nodes, and the scale bar represents the number of nucleotide substitutions per site. Presented are: (*a*) an overall tree with all *Tympanocryptis* species and outgroups included in analyses; (*b*) an expanded sub-tree of the grassland earless dragons; and (*c*) a distribution map of the grassland earless dragons overlain on estimated tussock and hummock grassland habitats [39].

representatives were available (Melbourne, VIC – *T. pinguicolla*; and Bathurst, NSW) and were found to be divergent from other eastern Australian earless dragons. The grassland earless dragons (GEDs) fall within this group. The GEDs [*T. lineata* (Canberra), *T. pinguicolla*, species A and species B] received strong phylogenetic support for monophyly (figure 2*b*), with 3.2% to 6.3% uncorrected genetic distance between lineages within the target group (the *T. lineata* grassland earless dragons (table 1*a*)).

In addition to the above analysis of all samples, we undertook a more limited phylogenetic analysis (using Bayesian inference) of 11 grassland earless dragon samples and including *T. houstoni* as an outgroup. We used only the 159 bp segment of the mtDNA ND2 protein coding gene that was sequenced from a long frozen specimen of species B for this comparison. Our analysis produced the same well-supported clades as those resolved in the full dataset, with *T. lineata* (Canberra) and species A being well-supported lineages that are not each other's sister (figure 3). Overall, there was 3.7% to 7.5% uncorrected genetic distances between lineages for this small region of ND2 (table 1*b*). The main difference between the full and reduced (159 bp fragment) analyses was that *T. lineata* (Canberra) is moderately supported as the basal group among the grassland earless dragons (posterior probability 85%) in the reduced analysis (figure 3) but not when all data were used.

Based on our phylogenetic results we investigated delimitation of putative species in the temperate grassland earless dragon group, including *T. lineata* (Canberra), species A (Monaro), species B (Bathurst), and *T. pinguicolla* (Melbourne), using phylogenomics, external morphology and cranial geometric morphometrics.

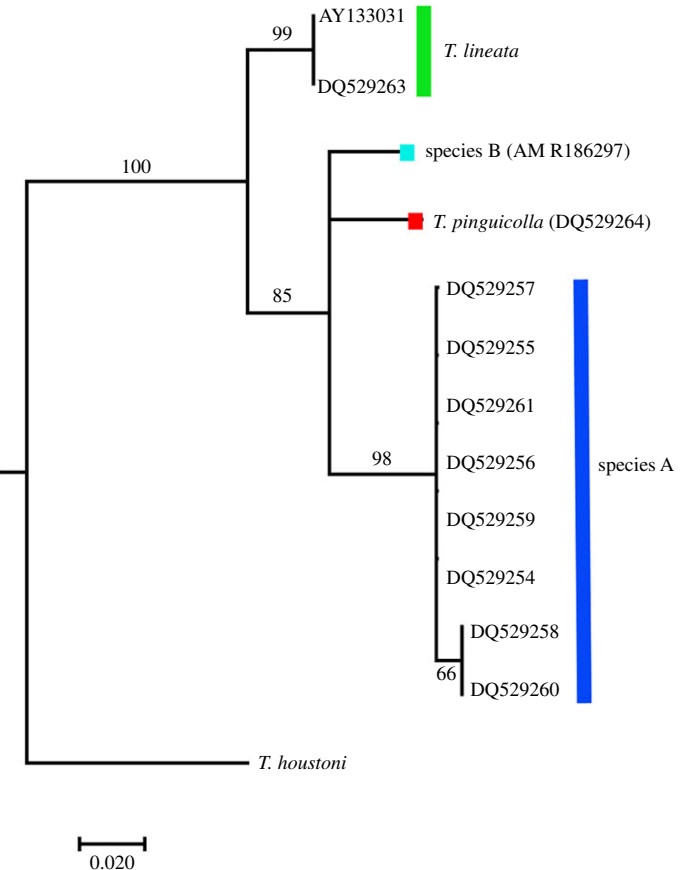

**Figure 3.** Bayesian phylogeny of described and putative grassland earless dragons using approximately 160 bp fragment of the *ND2* (NADH dehydrogenase subunit two) protein coding gene. Posterior probabilities are indicated on nodes, and the scale bar represents the number of nucleotide substitutions per site.

**Table 1.** Range (min. – max.) of pairwise uncorrected genetic distances between described and putative *Tympanocryptis* species in the *T. lineata* grassland earless dragons: (*a*) based on approximately 1800 bp of the mtDNA sequence including the gene *ND2* (NADH dehydrogenase subunit two) and flanking genes encoding tRNA$^{Trp}$, tRNA$^{Ala}$, tRNA$^{Asn}$, tRNA$^{Cys}$, tRNA$^{Tyr}$; and (*b*) the 159 bp fragment of the *ND2* gene that was successfully sequenced for the frozen specimen of species B.

|  | *T. lineata* | species A | species B |
|---|---|---|---|
| (*a*) | | | |
| species A | 0.032 – 0.039 | . | |
| species B | 0.061 | 0.049 – 0.056 | . |
| *T. pinguicolla* | 0.051 – 0.052 | 0.039 – 0.052 | 0.063 |
| (*b*) | | | |
| species A | 0.069 – 0.075 | . | |
| species B | 0.061 | 0.050 – 0.057 | . |
| *T. pinguicolla* | 0.064 | 0.037 – 0.045 | 0.061 |

### 3.2.2. Phylogenomics

A total of 95 individuals and 6930 SNP loci were included in the analyses at the completion of filtering covering the current distributions for *T. lineata* and species A. *Tympanocryptis pinguicolla* and species B were not included in the genomic assay as we were only able to extract and amplify small fragments of mtDNA. The PCA of the 6930 SNP loci shows clear geographical-based genomic structure between *T. lineata* (Canberra) and species A (Monaro) (figure 4) and is consistent with results previously found

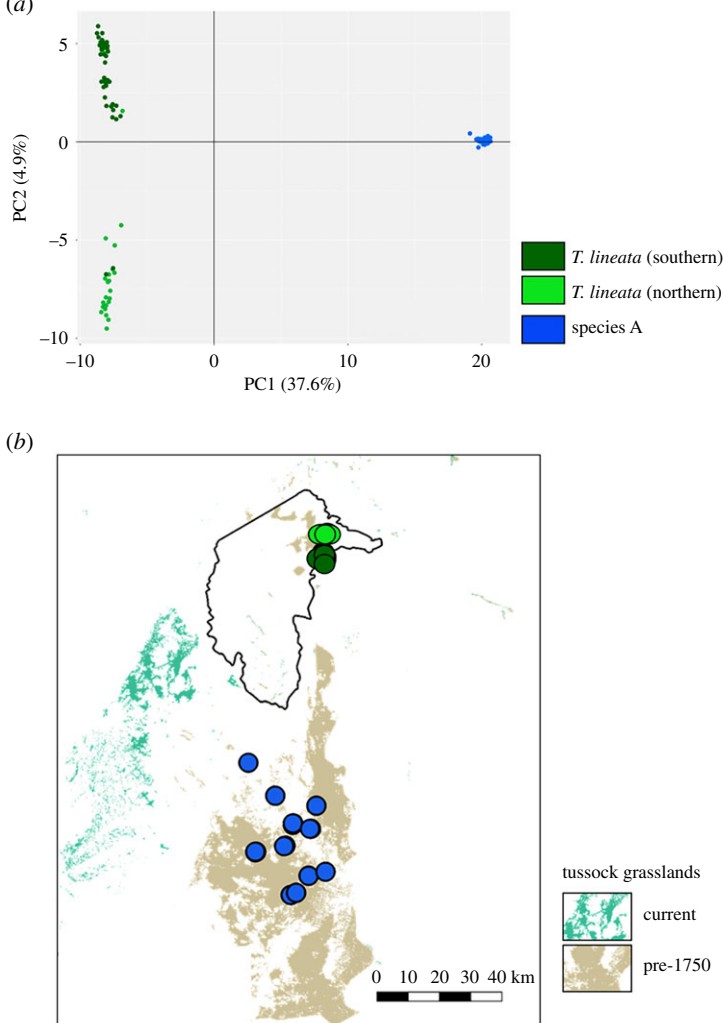

**Figure 4.** Phylogenomics of putative grassland earless dragon species in northern and southern Canberra (*T. lineata*) and Cooma (species A) using 6930 SNP loci: (*a*) principle coordinates analysis; and (*b*) distribution map overlain on estimated current and pre-1750 tussock grasslands [39,41].

using microsatellite DNA [42]). That same clear structuring is reflected in high levels of $G'_{ST}$ (0.471) and moderate Jost's D (0.098) estimates and in the highly significant pairwise $F_{ST}$ values of 0.50–0.53 between *T. lineata* (Canberra) and species A (table 2). The total heterozygosity (0.180) was substantially higher than the estimated heterozygosity when including substructure (0.122), further indicating strong genetic structure between *T. lineata* (Canberra) and species A. Similarly, Mantel's statistic for isolation by distance was very high (but non-significant; $r = 0.9977$, $p = 0.1667$). Our data also indicate genetic structuring within the Canberra population into groups north and south of the Molonglo River but some gene flow between these two groups is also evident ($F_{ST} = 0.076$, $p < 0.001$; figure 4).

### 3.2.3. External morphology

Visual inspection of external morphology of putative species revealed clear differences in dorsal scalation, with *T. lineata* and species A having very spinous enlarged dorsal scales, which are distinctly longer than wide with the posterior strongly convex and without a raised edge (figure 5). There is also strong individual variation in the size and number of enlarged spinous dorsal scales in these two species, from barely enlarged scales to numerous strongly enlarged. This individual variation does not appear to be related to the sex of the animal but is possibly related to size and maturity, with most of the animals with very small enlarged spinous dorsal scales being smaller animals. Further investigation using larger sample sizes is required to test for a relationship between

(*a*) *T. pinguicolla*      (*b*) *T. lineata* ZMB 740

(*c*) *T. lineata* Canberra

(*d*) species A Monaro

(*e*) species B Bathurst

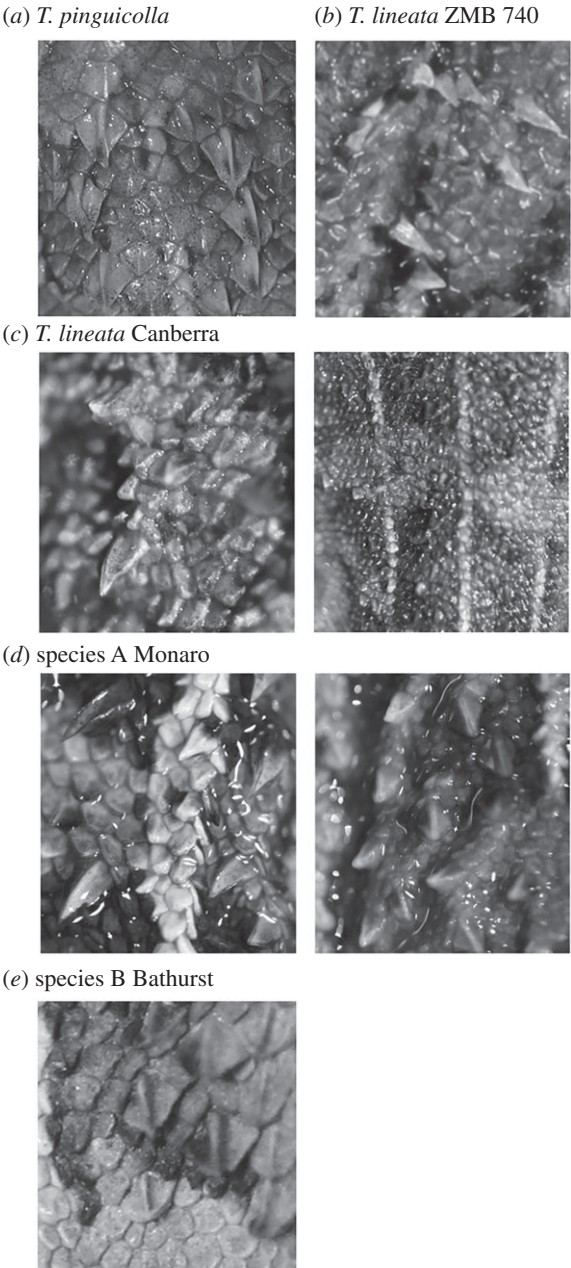

**Figure 5.** Detail of enlarged dorsal scales in described and putative species of grassland earless dragons: (*a*) *T. pinguicolla*; (*b*) *T. lineata* Lectotype ZMB 740; (*c*) *T. lineata* showing level of variation between individuals; (*d*) species A, Monaro, showing level of variation between individuals; and (*e*) species B, Bathurst.

**Table 2.** Pairwise $F_{ST}$, based on phylogenomic SNP analysis, of *T. lineata* populations in Canberra and species A (Monaro) above the diagonal, supported by *p*-values below the diagonal.

|  | *T. lineata* (southern Canberra) | *T. lineata* (northern Canberra) | species A |
|---|---|---|---|
| *T. lineata* (southern Canberra) | — | 0.075844 | 0.502909 |
| *T. lineata* (northern Canberra) | <0.0001 | — | 0.527242 |
| species A | <0.0001 | <0.0001 | — |

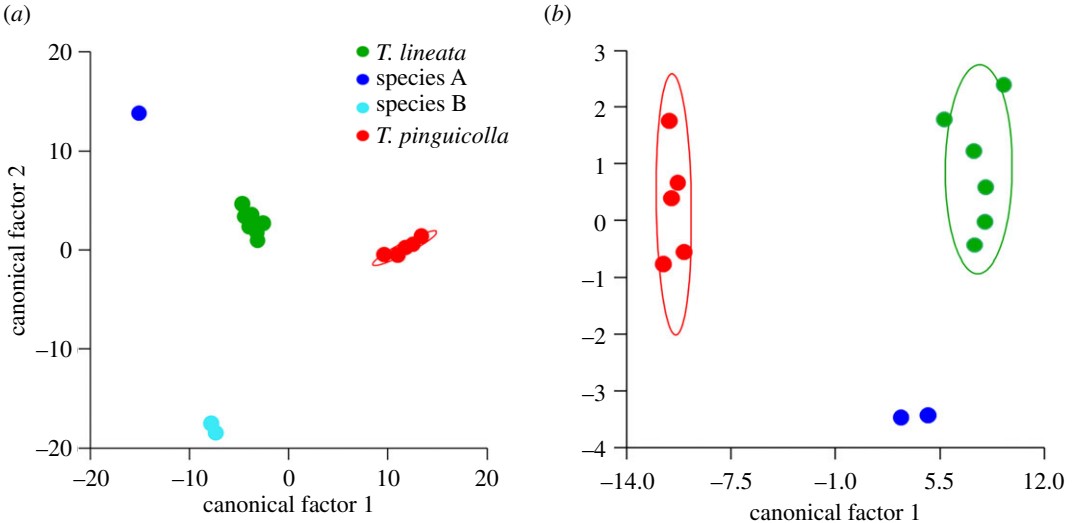

**Figure 6.** Discriminant functional analyses of (*a*) males and (*b*) females of the two described and two putative grassland earless dragon species, using eleven external morphological measurements (table 5).

age and enlargement of the spinous dorsal scales. In contrast to these two species, species B has enlarged dorsal scales that are not highly spinous, are as wide as long, and have a narrowly enlarged posterior edge (figure 5) while *T. pinguicolla* has enlarged spinose scales more similar to those of *T. lineata* and species A but tending to have a less convex and a narrowly raised posterior edge (figure 5). In addition, ventral scales differ between putative species with species B having weakly keeled scales, while those of the other species are smooth.

The DFA among the four GEDs was not able to distinguish groups based on external morphology for either males (Wilks' $\lambda_{12,13,12} = 0.000$, $F_{36,2} = 1.522$, $p = 0.476$) or females (Wilks' $\lambda_{10,2,10} = 0.002$, $F_{20,2} = 1.909$, $p = 0.40$). That lack of significance may well be an effect of small sample sizes, as the DFA correctly classified 100% of males and females into the assumed *a priori* species classes (figure 6). In males, canonical factors 1 (55.4% of variance) and 2 (44.1% of variance), when corrected for within-group variance, were associated most with tail length and forelimb length with male *T. pinguicolla* having longer tails and shorter forelimbs than the other species while species B had shorter tails and longer forelimbs than other species. In females, canonical factors 1 (97.1% of variance) and 2 (2.2% of variance), when corrected for within-group variance, were associated most with tail length and neck width. Female *T. pinguicolla* had longer tails and narrower necks than *T. lineata* and A, while there were no female species B available.

### 3.2.4. Geometric morphometrics

There were significant but low effects of sex ($R^2 = 0.091$, $p < 0.0001$) and size ($R^2 = 0.096$, $p < 0.0001$) on cranial morphology, and pairwise tests also indicated a significant but moderately low overall effect of species ($R^2 = 0.5048$, $p < 0.0001$) on cranial morphology with differences between all species, except species B and both *T. lineata* (Canberra) and species A (table 3). There were also distinct species-based groupings in the PCA (figure 7), with PC1 (associated with posterior and dorso-posterior cranium elongation and width and explaining 15.0% of variation), and PC2 (associated with central cranium elongation and width (influenced most by regions including the orbit, snout and dorsal areas) explaining 10.7% of variation.

### 3.2.5. Basis of taxonomic decisions

The mtDNA phylogeny of the *Tympanocryptis lineata* group, including all previously described subspecies, demonstrated that this group is paraphyletic and that there is far greater diversity than is presently recognized. We identified the five existing named taxa and eight additional unnamed species (figure 2). Below, we address the taxonomic status of the four putative species of grassland earless dragon as identified above. Using an integrative taxonomic (ITAX) approach, which follows the principle that as many lines of evidence as available should be combined to delimit species, we used criteria for assessment outlined in Melville *et al.* [7] where at least two lines of independent

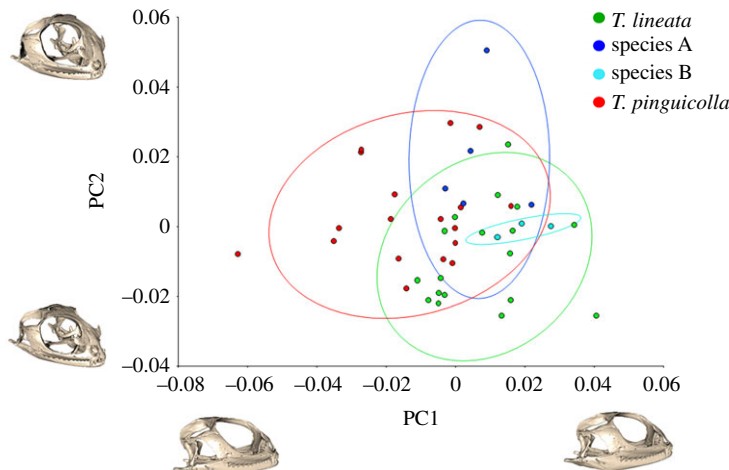

**Figure 7.** Principle components analysis of cranial morphology variation between grassland earless dragons, with 90% confidence ellipses shown, with warped graphical representations of the cranium morphology at each principle component's axis minimum and maximum value provided adjacent to their respective axis, in the perspective demonstrating the greatest variation.

**Table 3.** Pairwise tests of cranial morphology variation from geometric morphometric analysis of described and putative *Tympanocryptis* species, with least-squares mean distances above the diagonal and *p*-values below the diagonal. Values in italics indicate significance.

|  | *T. pinguicolla* | *T. lineata* | species A | species B |
|---|---|---|---|---|
| T. pinguicolla | — | *0.0317* | *0.0412* | *0.0399* |
| T. lineata | *0.0001* | — | *0.0379* | 0.0421 |
| species A | *0.0001* | *0.0012* | — | 1.1449 |
| species B | *0.0432* | 0.4294 | 0.1285 | — |

**Table 4.** Summary of characters useful for distinguishing among species of the '*T. lineata*' grassland earless dragons. Less common character states in bold.

| character | *T. pinguicolla* | *T. lineata* | *T. osbornei* | *T. mccartneyi* |
|---|---|---|---|---|
| spinose scales on thighs | yes | **no** | **no** | yes |
| ventral body scales | smooth | smooth | smooth | **weakly keeled** |
| throat scales | smooth | smooth | smooth | **keeled** |
| number of dark dorsal crossbands | 5–7 | 6–7 | 6–7 | **5** |
| prominent pale supraocular bar | present | **weak** | **weak** | present |
| belly with extensive black patterning | variable | variable | variable | **no** |
| number of caudal blotches | 12–14 | **7–12** | 12–14 | 12–14 |

evidence were required to delimit taxa. Evidence based on phylogenetic (mtDNA) analysis, genomics (*T. lineata* and species A), external morphology and geometric morphometric analysis of micro X-ray CT scans support four species of GEDs in southeastern Australia.

Grassland earless dragons are tussock grassland specialists and have been geographically separated historically, occurring only in the fragmented temperate tussock grasslands [40]. Even the geographically closest species, Canberra (*T. lineata*) and Monaro (species A), have been historically isolated [40]. It is estimated that the distribution of natural temperate grassland pre-European settlement was patchy and fragmented, acting to isolate these species that are both highly specialized to grasslands. *Tympanocryptis lineata* and species A are distinguished with multiple lines of evidence, including genomics (SNPs) and microsatellite DNA [42] that show no evidence of gene flow, mtDNA and external morphology. The

other two species in the GED clade, *T. pinguicolla* and species B (Bathurst), are both geographically isolated from each other and all other species but are also distinguished by the mtDNA phylogeny and morphology. In particular, species B (Bathurst) is morphologically divergent with clearly distinguishable dorsal scalation compared with the other GED species and multivariate analyses confirmed that this species is significantly morphologically distinguishable from the other taxa.

Based on these results we provide a taxonomic revision of the *T. lineata* grassland earless dragons, redefining two existing taxa (*T. lineata* and *T. pinguicolla*) and describing two new species (species A and species B). The other putative species identified in our phylogenetic analyses are the subjects of separate taxonomic study.

## 3.3. Taxonomic revision

*Tympanocryptis lineata* Peters, 1863.

Canberra Grassland Earless Dragon

Tables 4 and 5, figure 8.

Peters, W. 1863. *Monatsberichte der Königlichen Preussischen Akademie der Wissenschaften zu Berlin* 1863: 228–236 [1864 on title page] [230]. Lectotype ZMB 740, 'New Holland' [vicinity of Canberra, Australian Capital Territory]. Collector Lhotzki [Dr John Lhotsky].

*Tympanocryptis telecom* Wells, R. & Wellington, C.R. 1985. *Australian Journal of Herpetology*, Supplementary Series: 1–61 [20]. Type locality, Black Mountain, Australian Capital Territory. Type specimen not identified, *nomen nudum*.

Diagnosis. A species of *Tympanocryptis* with tapering snout, nasal scale below the canthus rostralis, six or seven dark dorsal crossbands, lateral skin fold, dorsal tubercles terminating in a prominent spine directed posterodorsally, lacking tubercular scales on the thighs, smooth gular scales, frequent presence of dark speckling on the ventral surfaces, especially the throat, and with 11 or fewer caudal blotches.

Description. Lateral neck fold well developed, from angle of jaw to gular fold; spines along extent of fold. Head and snout with strongly keeled dorsal scales; keels irregular, those on the lateral scales aligned more obliquely than those on the more medial scales. Snout shape smoothly tapering in profile, the canthal scales continuous with the rostral scale. Nasal scale dorsal margin does not cross onto the dorsal side of the *canthus rostralis*. No row of enlarged scales along the ventral margin of the nasal scale between the nasal and small snout scales. Dorsal body scales weakly to moderately keeled and imbricate. Numerous scattered strongly enlarged spinous dorsal scales, at least twice the width of adjacent body scales, each with a strong median keel ending in a prominent spine directed posterodorsally; sharply convex trailing edge not raised into a rim. Ventral body scales and throat scales smooth. Thigh scalation homogeneous, lacking scattered enlarged tubercular scales. Lateral fold between axilla and groin present. Snout–vent length 44–61 mm; femoral pores = 0; preanal pores = 2.

Dorsal colour pattern variable in degree of development and colour hue, from light brown to grey-brown with six or seven dark brown transverse bands and with 5-lined pattern well-defined, and usually continuous, or at most briefly interrupted on the paler interspaces between the dark cross bands. Dorsolateral lines as wide as or wider than the vertebral line, well defined, straight edged, not expanding around the vertebral blotches. Vertebral and dorsolateral stripes continue weakly onto the tail, outlining 7–11 dark caudal blotches. Pale supra-ocular bar present but usually weakly contrasting. Venter whitish, often heavily patterned with blackish speckling, especially on the throat.

Variation. Some individual variation in dorsal tubercule scales, from strongly spinose, to weakly spinose. The cause of the strong individual variation in the size and shape of the enlarged scales on the dorsal surface is unclear. It does not appear to be a sexually dimorphic trait with both males and females showing variation in spinose scales. The level of ventral mottling is variable between individuals, with some individuals having little or no dark pigmentation. However, animals frequently have extensive mottling across ventral surfaces, including head, throat, body, tail base and legs. Both males and females can exhibit orange-pink flush on ventral surface of their body and limbs.

Comparison to other species. With a distribution restricted to grasslands around Canberra, *T. lineata* is geographically isolated and does not overlap with any other *Tympanocryptis* species. *Tympanocryptis osbornei* sp. nov., occurring on the Monaro high plains in NSW is the geographically closest (approx. 100 km) and the two species are very similar in external morphological characters but show little overlap in morphometrics. An external character that assists in separating the two species is a trend for fewer caudal blotches in *T. lineata*, usually 7–11 versus 12–14 in *T. osbornei* sp. nov.

Distribution. Grassland around Canberra. Anecdotally considered locally common in the Canberra region before 1970, being found in what are now central areas of Canberra city [15]. Since surveys

**Table 5.** Morphological character measurements of 'T. lineata' grassland earless dragons. Mean ± standard error provided for measurements, range provided for counts (SubL). Character abbreviations: sub-digital lamellae (SubL); snout−vent length (SVL); tail length (TL); snout width (SnW); head width (HdW); neck width (NW); head length (Hdl); interlimb length (IntL); forelimb length (FL); and hindlimb length (HL).

| species | N | SubL | SVL | TL | SnW | HdW | NW | HdL | HdD | IntL | FL | HL |
|---|---|---|---|---|---|---|---|---|---|---|---|---|
| *T. lineata* | | | | | | | | | | | | |
| Lectotype ZMB 740 (female) | — | 18 | 57 | 68 | 4.8 | 11.1 | 5.7 | 16.5 | 8.7 | 24.5 | 23.3 | 33.0 |
| male | 8 | 17–22 | 51.8 ± 2.16 | 73.3 ± 3.40 | 4.9 ± 0.10 | 11.7 ± 0.31 | 4.0 ± 0.5 | 18.9 ± 0.81 | 8.6 ± 0.36 | 22.2 ± 0.86 | 24.1 ± 0.93 | 38.1 ± 1.16 |
| female | 6 | 18–21 | 50.7 ± 0.48 | 63.8 ± 4.25 | 4.7 ± 0.15 | 11.0 ± 0.28 | 3.7 ± 0.52 | 17.9 ± 0.80 | 8.3 ± 0.28 | 22.5 ± 1.39 | 21.73 ± 1.27 | 34.7 ± 1.98 |
| *T. pinguicolla* | | | | | | | | | | | | |
| Holotype SAMA R2468a (female) | — | 19 | 62 | 71 | 5.8 | 12.5 | 6.1 | 17.1 | 9.6 | 31.9 | 27.6 | 39.3 |
| male | 6 | 18–22 | 50.1 ± 2.56 | 60.2 ± 2.50 | 5.1 ± 0.23 | 11.5 ± 0.48 | 3.8 ± 0.28 | 18.5 ± 0.79 | 8.9 ± 0.36 | 22.1 ± 1.30 | 21.6 ± 0.88 | 35.1 ± 1.14 |
| female | 6 | 17–20 | 54.2 ± 3.84 | 61.3 ± 3.84 | 5.17 ± 0.29 | 11.9 ± 0.48 | 4.4 ± 0.36 | 18.2 ± 0.86 | 8.8 ± 0.48 | 26.0 ± 2.97 | 22.7 ± 1.26 | 35.1 ± 1.46 |
| *T. osbornei* (species A) | | | | | | | | | | | | |
| Holotype SAMA R43098 (male) | — | 21 | 49 | 73 | 11.1 | 4.2 | 3.4 | 18.8 | 7.2 | 20.3 | 23.0 | 35.8 |
| male | 2 | 19–21 | 53.5 ± 4.5 | 70.0 ± 3.0 | 4.6 ± 0.34 | 12.3 ± 1.15 | 3.9 ± 0.55 | 19.4 ± 0.65 | 8.4 ± 1.21 | 22.3 ± 1.93 | 25.2 ± 2.16 | 37.4 ± 1.64 |
| female | 2 | 19–21 | 44.0 ± 0.0 | 49.5 ± 2.5 | 4.6 ± 0.15 | 10.4 ± 0.07 | 3.5 ± 0.09 | 15.7 ± 0.03 | 7.17 ± 0.25 | 18.1 ± 1.38 | 19.82 ± 0.37 | 30.98 ± 0.48 |
| *T. mccartneyi* (species B) | | | | | | | | | | | | |
| Holotype AM R26077 (male) | — | 23 | 53 | 75 | 7.3 | 10.9 | 7.5 | 16.5 | 8.7 | 22.1 | 20.2 | 35.3 |
| male | 2 | 22–23 | 52 ± 1.0 | 75 ± 0.0 | 6.9 ± 0.20 | 11.0 ± 0.16 | 7.0 ± 0.20 | 16.4 ± 0.06 | 8.0 ± 0.68 | 22.2 ± 0.14 | 21.8 ± 1.57 | 35.6 ± 0.35 |
| female | 0 | — | — | — | — | — | — | — | — | — | — | — |

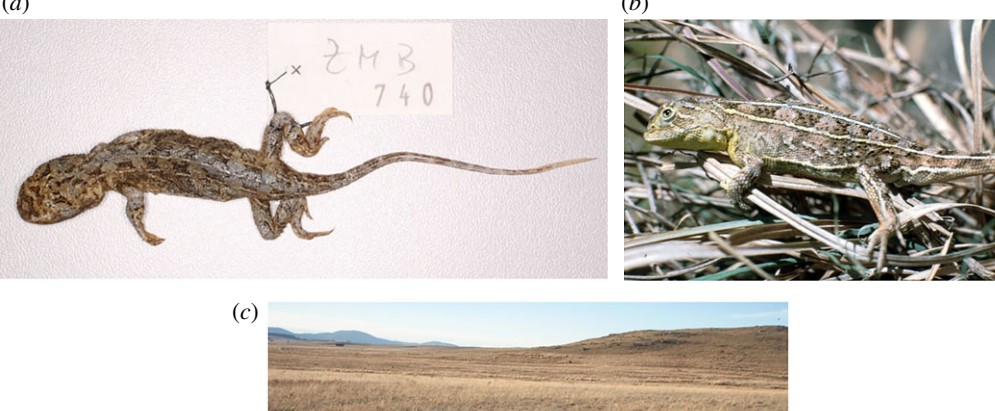

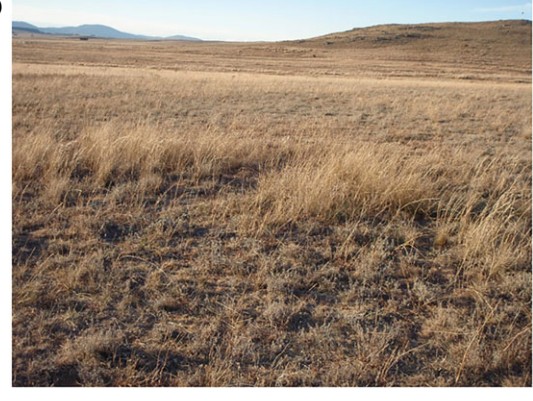

**Figure 8.** *Tympanocryptis lineata*: (*a*) Lectotype ZMB 740; (*b*) Adult male, Majura, Australian Capital Territory, photo taken in 1992 (W. Osborne); and (*c*) Jerra East, Canberra, tussock grassland habitat (photo: L. Doucette).

were initiated in 1997, populations of the species have been detected at 10 sites, all within 13 km of the Canberra International Airport [43]. The population sizes at several of those sites have declined since surveys began and the lizards have become undetectable in some places [43]. The populations that remain are heavily fragmented by roads, airports and other infrastructure forming genetically discrete populations in most cases [44]. A major population distributional disjunction that may pre-date the European settlement of Canberra occurs to the north and south of the Molonglo River which splits Canberra.

Habitat. Characterized by open-structured tussock grasslands with few or no trees and shrubs [11,12], limited or no fertilization or pasture improvement, and comprising slightly higher ground in well-drained areas [11]. The lizards rely on the existence of arthropod burrows to survive the extreme (low and high) temperatures experienced in these grasslands (L Doucette *et al.*, unpublished data).

Remarks. Populations at those sites that are routinely monitored have remained low but stable (with some fluctuations) or have declined to very low levels since population monitoring began in 2002 (Stringer *et al.* unpublished data). At some sites, the dragons have gone below detectable levels [43], with more survey work required to determine whether they have been truly extirpated at those sites. This research has found that there was a gradual non-significant decline in population size of *T. lineata* from 1995 to 2006 and then dramatic reductions (88% decline at the most densely populated site). Dimond *et al.* estimated a low annual survival rate for individuals, both in juvenile and adult lizards. Based on their findings, the authors suggest that this species is at high risk of extinction [43]. In addition, population genetic work, based on microsatellites, has found that there are separate conservation units within the Canberra region and suggest that development or contraction of the grasslands is likely to enhance fragmentation, reduce connectivity and hasten the extinction of this species [42].

*Tympanocryptis pinguicolla* Mitchell, 1948
Victorian Grassland Earless Dragon
Tables 4 and 5, figure 9.
*Tympanocryptis lineata pinguicolla* Mitchell, F.J. 1948. *Records of the South Australian Museum* (*Adelaide*) 9: 57–86 [70, fig. 6 pl. 4]. Holotype, SAMA R2468a, 'Southern Victoria'.

Diagnosis. A species of *Tympanocryptis* with tapering snout, nasal scale below the canthus rostralis, 5–7 dark dorsal crossbands, lateral skin fold, dorsal tubercles with reduced development of a small vertically oriented terminal spine in only the largest individuals, heterogeneous thigh scalation

(a)

(b)

(c)

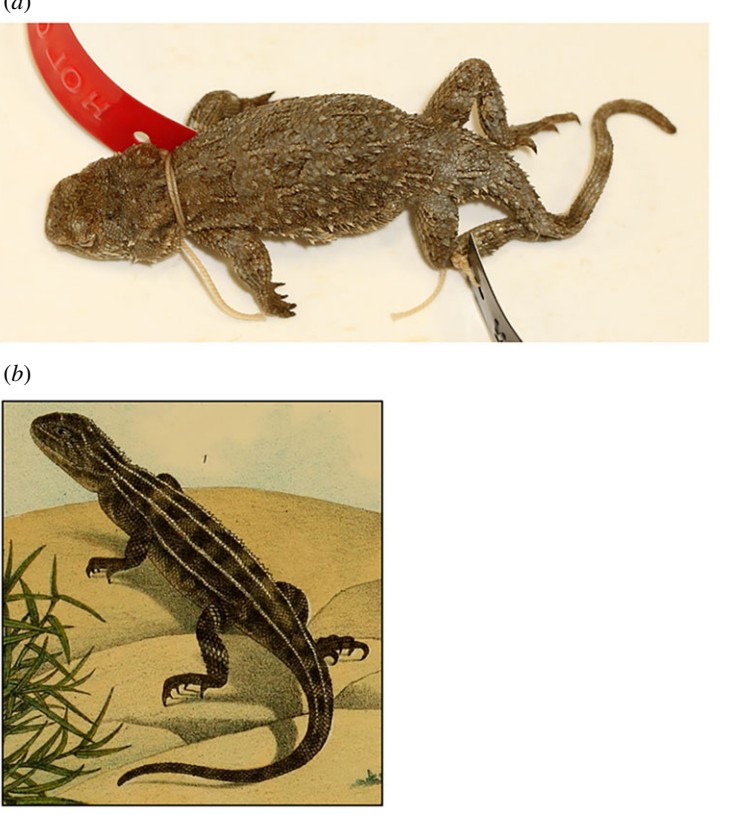

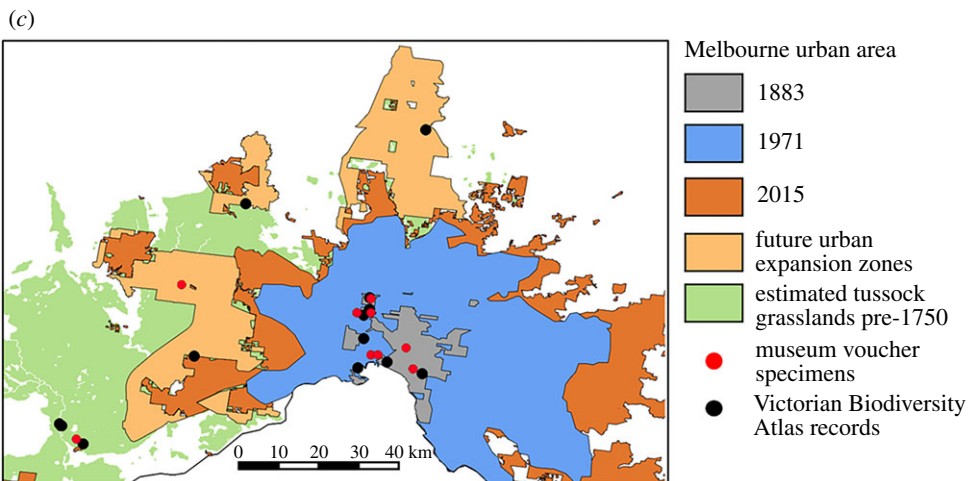

**Figure 9.** *Tympanocryptis pinguicolla:* (*a*) Holotype SAMA R2468a; (*b*) illustration from [45]; and (*c*) distribution map of all *T. pinguicolla* samples (museum vouchers) and sighting locations (Victorian Biodiversity Atlas) overlain on historical boundaries of Melbourne urban area [46] and pre-1750 tussock grassland [41].

including scattered enlarged tubercles, smooth gular scales, frequent presences of dark speckling on the ventral surfaces, especially the throat.

Description. Lateral neck fold well developed, from angle of jaw to gular fold; spines along extent of fold. Head and snout with strongly keeled dorsal scales; keels irregular, those on the lateral scales aligned more obliquely than those on the more medial scales. Snout shape smoothly tapering in profile, the canthal scales continuous with the rostral scale. Nasal scale dorsal margin does not cross onto the dorsal side of the *canthus rostralis*. No row of enlarged scales along the ventral margin of the nasal scale between the nasal and small snout scales. Dorsal body scales weakly to moderately keeled and imbricate. Numerous scattered strongly enlarged spinous dorsal scales, at least twice the width of adjacent body scales, each with a strong median keel, terminal spine only in the larger individuals; apex of spinose scales directed almost vertically; sharply convex trailing edge not raised into a rim.

Ventral body scales and throat scales smooth. Thigh scalation heterogeneous, with scattered enlarged tubercular scales similar to those on body. Lateral fold between axilla and groin present. Snout–vent length 45–62 mm; femoral pores = 0; preanal pores = 2.

Dorsal colour pattern variable in degree of development and colour hue, from reddish brown to grey-brown, usually with five, but up to seven dark brown transverse bands and with 5-lined pattern well defined, and usually continuous, or at most briefly interrupted on the paler interspaces between the dark cross bands. Dorsolateral lines as wide as or wider than the vertebral line, well defined, straight-edged, not expanding around the vertebral blotches. Vertebral and dorsolateral stripes continue weakly onto the tail outlining 12–14 dark caudal blotches. Pale supra-ocular bar usually strongly contrasting. Venter whitish, often heavily patterned with blackish speckling, especially on the throat.

Variation. Variation between individuals not pronounced, with patterning, coloration and scalation fairly consistent. The lateral fold between the axilla and groin, with associated pale enlarged scales, can be weak in some individuals. For example, the Holotype SAMA R2468a lacks a lateral stripe and has a weak skin fold between the axilla and groin. The width of the neck is also variable between individuals but is always as wide or wider than the head. It was suggested in the original species description that the width of the neck is dependent on fat storage for over-wintering at southern latitudes [47]. A photograph by Frank Collett (reproduced in Jenkins & Bartell [48]) shows an adult male *T. pinguicolla* from Little River, Victoria, with bright yellow colouring on the throat and orange on the lateral margins of the belly.

Comparison to other species. *Tympanocryptis pinguicolla*, with a distribution restricted to grasslands on the Victorian basalt plains around Melbourne, is geographically isolated and does not occur in close proximity to any other *Tympanocryptis* species. *Tympanocryptis lineata*, and *T. osbornei* sp. nov. are geographically closest (greater than 300 km). *Tympanocryptis pinguicolla* can be distinguished from all other *Tympanocryptis* in its almost vertically oriented dorsal tubercles that either lack a terminal spine or have only a small projection. They further differ from the two southern highlands species (*T. lineata* and *T. osbornei*) in the presence of enlarged tubercular scales scattered on the thighs, compared to the absence of this scalation, and differ from the currently undescribed populations of *Tympanocryptis* in northwest Victoria and adjacent South Australia in frequently having six or seven transverse dark dorsal bands (versus never more than five) and in the presence (versus absence) of a lateral skin fold.

Distribution. Grassland on the Victorian basalt plains around Melbourne (figure 9). Historically known from several locations around Melbourne, including Sunbury, Maribyrnong River (called 'Saltwater River') and as far west as the Geelong area [45,47,49–51] up until the late 1960s. Other old records are from Rutherglen and Maryborough in central Victoria [49]. More recently, six sightings have been reported between 1988 and 1990: Craigieburn (one recorded sighting), Merri Creek (one recorded sighting), Holden Flora Reserve (one recorded sighting), and Little River, west of Werribee (three recorded sightings). Intensive trapping surveys at these three locations immediately following these sightings and in subsequent years have failed to confirm the presence of *T. pinguicolla* [15,52].

Habitat. A specialist species inhabiting native temperate grasslands in Victoria. Although there is little information available about the ecology and habitats of this species, it was described by Lucas & Frost in 1894 [49] as:

> Inhabiting stony plains and retreating into small holes, like those of the 'Trap-door Spider,' in the ground when alarmed. (McCoy, l.c.). Often met with under loose basalt boulders.

Based on this description, it might be assumed that the ecology and habit of this species may be similar to the earless dragons from the ACT and Cooma, NSW, regions, for which habitat requirements and ecology have been studied.

Remarks. *Tympanocryptis pinguicolla* is listed as Endangered at a federal level, with a National Recovery Plan that is a decade old [15] and is currently listed as critically endangered on the Advisory List of Threatened Vertebrate Fauna in Victoria [53]. The last confirmed sighting of the grassland earless dragon in Victoria was from the Geelong area in July 1969 [51]. Extensive surveys have been undertaken on *T. pinguicolla* (detailed in [15]) around Melbourne and Geelong, with recent intensive efforts by researchers at Zoos Victoria [52]. However, there are still potential habitat areas which are yet to be intensively surveyed [52]. With this taxonomic revision, it is now 50 years since this species has been seen alive and, given the considerable but unsuccessful efforts noted above to locate the species [15,52], raises the possibility that this species is extinct. Below, we estimate the chance of extinction using sighting data. Continued targeted survey efforts in remaining suitable habitats [52], particularly those regions not yet surveyed, is of utmost importance and a priority for conservation management, to confidently determine the status of this species.

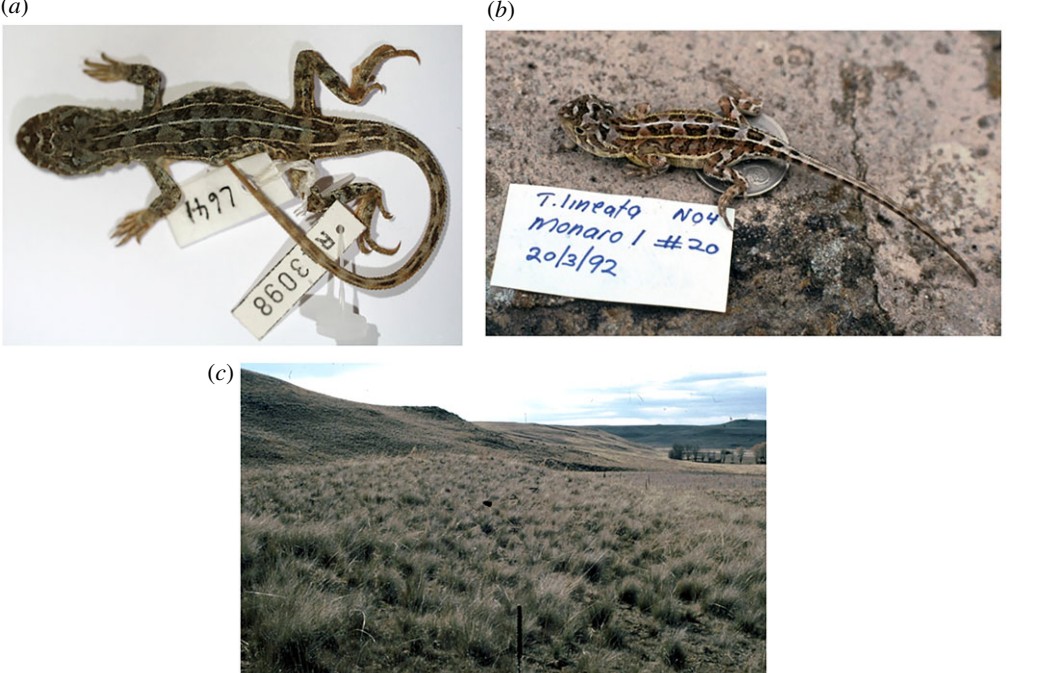

**Figure 10.** *Tympanocryptis osbornei* sp. nov.: (*a*) Holotype SAMA R43098; (*b*) Male, Monaro, New South Wales, photo taken in 1992 (W. Osborne); and (*c*) Monaro, New South Wales, basalt tussock grassland habitat ( photo: W. Osborne).

*Tympanocryptis osbornei* sp. nov.
ZooBank LSID: urn:lsid:zoobank.org:act:02D9C489-2C7A-43B0-A698-2DFA35B1D1BE
Monaro Grassland Earless Dragon
Tables 4 and 5, figure 10.
[Referred to as species A in results]

Holotype. SAMA R43098, Cooma, NSW. Adult male. [specific location details held in SAMA database but not released for publication due to conservation concerns].

Paratypes. NMV D76106, NMV D76105 Cooma, NSW. SAMA R43347, Cooma, NSW. AM R38172, R131836, Cooma, NSW.

Diagnosis. A species of *Tympanocryptis* with tapering snout, nasal scale below the canthus rostralis, six or seven dark dorsal crossbands, lateral skin fold, dorsal tubercles terminating in a prominent spine directed posterodorsally, lacking tubercular scales on the thighs, smooth gular scales, frequent presence of dark speckling on the ventral surfaces, especially the throat, and with 12 or more caudal blotches.

Description. Lateral neck fold well developed, from angle of jaw to gular fold; spines along extent of fold. Head and snout with strongly keeled dorsal scales; keels irregular, those on the lateral scales aligned more obliquely than those on the more medial scales. Snout shape smoothly tapering in profile, the canthal scales continuous with the rostral scale. Nasal scale dorsal margin does not cross onto the dorsal side of the *canthus rostralis*. No row of enlarged scales along the ventral margin of the nasal scale between the nasal and small snout scales. Dorsal body scales weakly to moderately keeled and imbricate. Numerous scattered strongly enlarged spinous dorsal scales, at least twice the width of adjacent body scales, each with a strong median keel ending in a prominent spine directed posterodorsally; sharply convex trailing edge not raised into a rim. Ventral body scales and throat scales smooth. Thigh scalation homogeneous, lacking scattered enlarged tubercular scales. Lateral fold between axilla and groin present. Snout–vent length 49–58 mm; femoral pores = 0; preanal pores = 2.

Dorsal colour pattern variable in degree of development and colour hue, from reddish brown to grey-brown with six or seven dark brown transverse bands and with 5-lined pattern well defined, and usually continuous, or at most briefly interrupted on the paler interspaces between the dark cross bands. Dorsolateral lines as wide as or wider than the vertebral line, well defined, straight-edged, not expanding around the vertebral blotches. Vertebral and dorsolateral stripes continue weakly onto the tail outlining 12–14 dark caudal blotches. Pale supra-ocular bar present but usually weakly contrasting. Venter whitish, often heavily patterned with blackish speckling, especially on the throat.

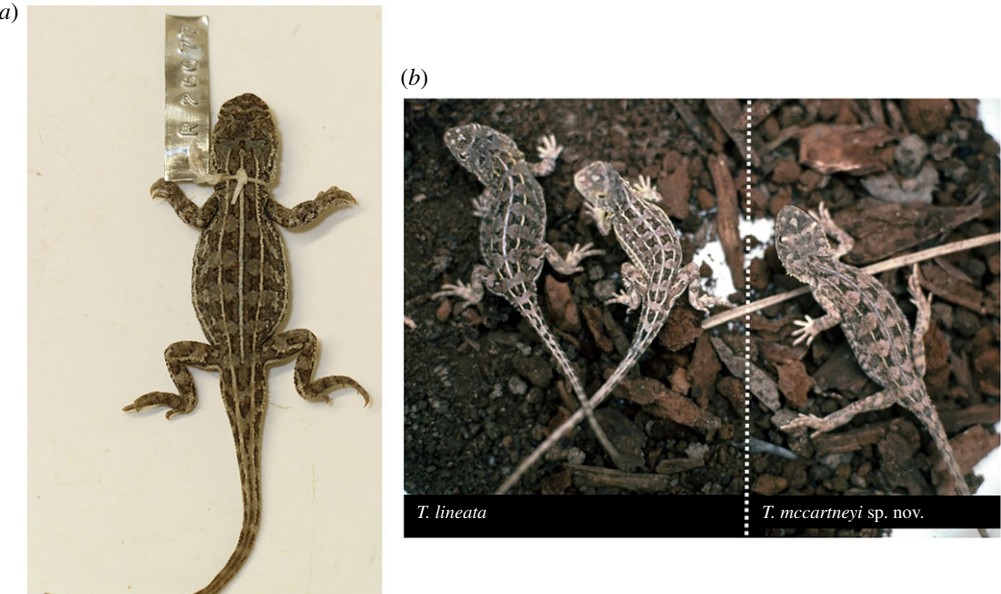

**Figure 11.** *Tympanocryptis mccartneyi* sp. nov.: (*a*) Holotype AM R26077; and (*b*) comparison of grassland earless dragons, left—female *T. lineata* from 'the Poplars', which was the first ACT population rediscovered, middle—male *T. lineata* from Majura, ACT, and right—*T. mccartneyi* sp. nov. from Bathurst, NSW, found by Gavin waters (photo: W. Osborne).

Variation. Some individual variation in dorsal tubercule scales, from strongly spinose, to weakly spinose. The cause of this individual variation in the size and shape of the enlarged scales on the dorsal surface is unclear. It does not appear to be a sexually dimorphic trait with both males and females showing variation in spinose scales. The level of ventral mottling is variable between individuals, with some individuals having little or no dark pigmentation. However, animals frequently have extensive mottling across ventral surfaces, including head, throat, body, tail base and legs. Both males and females can exhibit orange-pink flush on ventral surface of their body and limbs.

Comparison to other species. *Tympanocryptis osbornei*, with a distribution restricted to grasslands on the Monaro tablelands, is geographically isolated and does not overlap with any other *Tympanocryptis* species. See *T. lineata* for comments on separating these two species.

Distribution. Confined to the Monaro tablelands in a 20 × 60 km region bounded by the Maclaughlin and Murrumbidgee Rivers in the south and north, the Monaro Highway in the east and Berridale in the west [54].

Habitat. Naturally treeless native grassland communities from 758 to 1234 m above sea level on basalt geology and heavy clay soils and in predominately dry tussock grasslands of snow grasses (*Poa* spp.), wallaby grasses (*Austrodanthonia* spp.) and kangaroo grass (*Themeda triandra*) [54]. The species overwinters in crevices or burrows excavated by wolf spiders beneath surface rocks [12] but its ecology has been little studied.

Etymology. Named for Dr Will Osborne, conservation biologist and ecologist, who provided the first accounts of this species in the modern era and has spearheaded ecological and conservation research.

Remarks. There has been ongoing survey work undertaken on *T. osbornei*, with conservation, ecological and genetic findings [42]. Population declines and contractions have been recorded in this species [54]. Only a single site on the Monaro is reserved and the species is no longer detectable at that site [54], although the detection rate of this species is extremely low making accurate estimation of population declines particularly difficult. Results of these surveys highlight the difficulties of studying this species and the imperative of continued work to ensure its future.

*Tympanocryptis mccartneyi* sp. nov.
ZooBank LSID: urn:lsid:zoobank.org:act:8AC42566-2404-4DCC-8503-7D910E86E824
Bathurst Grassland Earless Dragon
Tables 4 and 5, figure 11.
[Referred to as species B in results]
Holotype. AM R26077, Bathurst, NSW [specific location details held in AM database but not released for publication due to conservation concerns]. Adult male. Collected by I. McCartney.
Paratype. AM R25980, open grassland, Bathurst, NSW.

Diagnosis. A species of *Tympanocryptis* with tapering snout, nasal scale below the canthus rostralis, six dark dorsal crossbands, lateral skin fold, dorsal tubercles terminating in a prominent spine directed posterodorsally, heterogeneous thigh scalation including scattered enlarged tubercles, keeled gular scales, frequent presence of dark speckling on the ventral surfaces, especially the throat, and with 12 or more caudal blotches.

Description. Lateral neck fold well developed, from angle of jaw to gular fold; spines along extent of fold. Head and snout with strongly keeled dorsal scales; keels irregular, those on the lateral scales aligned more obliquely than those on the more medial scales. Snout shape smoothly tapering in profile, the canthal scales continuous with the rostral scale. Nasal scale dorsal margin does not cross onto the dorsal side of the *canthus rostralis*. No row of enlarged scales along the ventral margin of the nasal scale between the nasal and small snout scales. Dorsal body scales weakly to moderately keeled and imbricate. Numerous scattered strongly enlarged spinous dorsal scales, at least twice the width of adjacent body scales, each with a strong median keel ending in prominent spine directed posterodorsally; posterior edge weakly raised, not convex. Ventral body scales weakly keeled, throat scales keeled. Thigh scalation heterogeneous, with scattered enlarged tubercular scales similar to those on body. Lateral fold between axilla and groin present. Snout–vent length of the two known specimens, 53 mm (holotype) and 51 mm (paratype); femoral pores = 0; preanal pores = 2.

Dorsal colour pattern (in preservative) brownish grey with six dark brown transverse bands and with 5-lined pattern well defined and continuous. Dorsolateral lines as wide as or wider than the vertebral line, well defined, straight edged, not expanding around the vertebral blotches. Vertebral and dorsolateral stripes continue weakly onto the tail outlining 12–14 dark caudal blotches. Pale supra-ocular bar usually strongly contrasting. Venter whitish, with dark speckled pigmentation on throat and sides of the belly.

Variation. With only two well-preserved specimens available, it is not possible to assess variation within this species. Both specimens are similar in scalation, coloration and patterning. There is, however, variation in the amount of pigmentation on the ventral surface, ranging from faint mottling restricted to head, throat and upper chest (AM R25980), through to dark mottling on head, throat and upper/lateral body (AM R26077). Both specimens are male, so the nature of any sexual dimorphism is unknown.

Comparison to other species. *Tympanocryptis mccartneyi* sp. nov., with a distribution restricted to grasslands around Bathurst, NSW, is geographically isolated and is not known to overlap with any other *Tympanocryptis* species. *Tympanocryptis lineata*, occurring in the Canberra grasslands, is geographically closest (approx. 250 km). *Tympanocryptis mccartneyi* differs from both *T. lineata* and *T. osbornei* in having enlarged tubercular scales scattered on the thighs and keeled rather than smooth throat scales. Differs from *T. pinguicolla* from Victoria in having more acutely pointed dorsal tubercles directed more posteriorly than vertically and keeled rather than smooth gular scales.

Distribution. Grasslands and open country on the alluvial plains around Bathurst.

Habitat. A grassland specialist, inhabiting treeless plains and open grasslands. Has been found along railway tracks, with weedy *Paspalum* grass thickets, and in vacant paddocks with tall pasture grass. Presumably similar habits to those of grassland earless dragons further south.

Etymology. Named for Ian McCartney, retired ranger and local natural historian of the Bathurst region, who, along with Gavin Waters, was instrumental in the discovery of this species. They also organized and participated in field surveys with the ACT Herpetological Association, the first of which was on 11 December 1988 [55].

## 3.4. Estimating chance of extinction for *Tympanocryptis pinguicolla*

An outcome of our taxonomic revision is that *T. pinguicolla* is now restricted to Victoria. This means that the last confident sighting was in 1969 and the last animal confirmed with collection of a voucher specimen was 1967. More recent observations in 1988 and 1990 are recorded in the Victorian Biodiversity Atlas but are not considered with high confidence because of lack of confirmation with intensive follow-up surveys at those localities [15]. For this reason we undertake separate analyses to estimate the possibility that *T. pinguicolla* is extinct in 2019 with two datasets: the first dataset including sightings and records of high confidence and the second with the additional sightings from 1988 and 1990 (electronic supplementary material, table S3). Results between the two datasets were broadly similar (table 6), with at least one of the estimators for both datasets suggesting *T. pinguicolla* is still extant. With a cut-off of $\alpha < 0.05$, which is equivalent to a 95% chance of extinction, the OLE, Solow [35], and Robson & Whitlock [56] methods estimated an extinction date ($\tau_E$) before 2019 for both datasets. However, the Solow [36] measure estimated that *T. pinguicolla* was currently extant for the dataset comprised of the high confidence sightings and museum vouchers, while the Strauss &

**Table 6.** Estimates for *T. pinguicolla* extinction dates ($\tau_E$). Those estimates that reached a cut-off of $\alpha < 0.05$ by 2019, equivalent to a 95% chance of extinction, are in italics. For those estimators where $\alpha < 0.05$ was not reached by 2019 the $\alpha$ estimate for 2019 is provided.

| | museum records | | all sightings | |
|---|---|---|---|---|
| | $\tau_E$ | $\alpha$ | $\tau_E$ | $\alpha$ |
| OLE | *1994* | — | *1998* | — |
| Strauss & Sadler [34] | *2010* | *0.040* | extant | 0.09 |
| Solow [35] | *1993* | *0.044* | *2011* | *0.045* |
| Solow [36] | extant | 0.016 | *2016* | *0.047* |
| Robson & Whitlock [56] | *1993* | *0.040* | *2009* | *0.050* |

Sadler [34] estimator suggested *T. pinguicolla* was currently extant for the dataset that also including lower confidence sightings. Thus, results from the estimation methods as to the chance of *T. pinguicolla* being extinct in 2019 are equivocal.

# 4. Discussion

Our study provides a comprehensive revision of the grassland earless dragons of southeastern Australia, providing a stable taxonomy on which to base conservation management decisions. Beyond the conservation management of this group (discussed in further detail below), our study reinforces important issues that have existed for some decades regarding the absence of scientific rigour in applying scientific nomenclature. We agree with Thomson *et al.* [2] that rigorous scientific-based taxonomic research is essential to the conservation of species. However, the case of *T. lineata* shows that the current nomenclatural code [57] does not assist in this endeavour as much as it could. By only recommending, as opposed to requiring, that actions to stabilize nomenclature are accompanied by sufficient data and due diligence, the ICZN forces substandard work to be treated with an undeserved level of respect. The essential scientific process of species discovery remains underpinned by nomenclatural regulations that explicitly eschew involving the quality of the science in the decision making process [58,59].

## 4.1. Status of *Tympanocryptis pinguicolla*

With our taxonomic revision of the GEDs, *T. pinguicolla* is now restricted to Victoria. As already detailed, the last confirmed sighting of this species was more than 50 years ago in 1969 [51] at Little River, south-west of Melbourne. Traditional thought was that if a species had not been seen for 50 years it could be declared extinct, however, there is now far more evidence required before such a drastic step should be taken. There have been numerous cases of a species returning from extinction after not being seen for decades. For example, another Victorian species that was thought to be extinct by the early 1900s, the Leadbeater's possum (*Gymnobelideus leadbeateri*), was rediscovered in 1961 and later declared endangered, with changes to forestry practices implemented to conserve it into the future [60]. If the Leadbeater's possum had not been prematurely declared extinct, it may have received protection and conservation management far earlier. The consequences of declaring a species extinct are significant [60].

There are a range of methods available to assess the likelihood of extinction for species, including those based on sighting records, as used in our study, as well as survey models that integrate the per cent of habitats surveyed, likelihood of detection and probability of correct identification [60]. The equivocal results from our assessment using sighting records, that gave a range greater than 25 years in likelihood of extinction estimates (from extinct in 1993 to currently extant), suggests that these estimators are not able to provide accurate results with the sighting data available for *T. pinguicolla*. More complex and detailed estimates involving survey locations and effort with known habitat requirements and biology [60] would be an optimal approach for this species.

Butchart *et al.* [61] proposed a continuum from high to low confidence of extinction, on a spectrum from Extinct to Critically Endangered (Possibly Extinct) to Critically Endangered, which is used by the

IUCN Redlist. Under this grading a species would not meet either of the higher extinction categories if any of the following criteria were met [61]:

1. Surveys have not covered all potential habitats or activity periods.
2. The species is difficult to detect.
3. There have been reasonably convincing recent local reports or unconfirmed sightings.
4. Suitable habitat remains in the known range of the species.

In the case of *T. pinguicolla*, at least three of these four criteria are met. Specifically, Banks *et al.* [52] have undertaken detailed surveys across potential habitats, but there are remaining areas that have not been surveyed, and this species is small, cryptic and difficult to detect. Thus, we suggest that *T. pinguicolla* does not yet meet the criteria for being considered either Extinct or Critically Endangered (Possibly Extinct). However, this needs to be assessed in detail beyond the scope of our paper. Currently, *T. pinguicolla* is listed as Endangered at a federal level, with a National Recovery Plan that is a decade old [15]. Thus, there is an immediate urgency for imperative funding of further surveys across remaining habitats and a complete revision of the conservation status of this species.

## 4.2. Conservation implications

Our study will significantly impact future conservation management of grassland earless dragons in Victoria, NSW and the ACT. Currently, the (now) four species of GEDs are contained under a single conservation Action Plan that is a decade old [15] and now requires updating. In addition, the conservation management plans for *T. lineata* [50] are now invalid as they refer to the unnamed species across semi-arid regions of South Australia, northwestern Victoria and western NSW. Thus, it is of immediate priority to reassess the conservation status of each of the four GED species.

A considerable level of research and conservation effort has been focused on three of the GED species: *T. lineata*, *T. pinguicolla* and *T. osbornei* sp. nov. In Victoria, conservation efforts have focused on targeted surveys to locate remnant populations of *T. pinguicolla* [52], while in the ACT and NSW long-term monitoring, captive breeding and ecological research has provided an understanding of the population status, ecological requirements and biology of both *T. lineata* and *T. osbornei* sp. nov. The conservation management of these species needs to be reviewed in light of them now being short-range habitat specialists with restricted distributions.

In contrast to the other three species, virtually nothing is known of *T. mccartneyi* sp. nov. from the Bathurst region. Apart from the collection of a few specimens and a limited number of sightings more than 20 years ago, we have virtually no understanding of this species. A priority for this species is adequate funding to undertake targeted survey work on suitable habitats, based on the expert knowledge of the local naturalists that discovered this species. Such survey work will be critical in assessing the distribution, habitat requirements and population health of this species. We believe that it would be worthwhile undertaking surveys across the higher altitude remnant native grasslands of the region, including Bathurst, Orange, Blayney and even south towards Cowra. These regions have all been listed under Natural Temperate Grassland of the South Eastern Highlands ecological communities, which were recently designated at the federal level as Critically Endangered [62]. These grasslands occur at altitudes up to approximately 1200 m and are naturally treeless or sparsely treed, with native tussock grasses being the dominant vegetation. These grasslands have declined by more than 90% and now occur as highly fragmented patches that are mostly less than 10 ha in size [62], with only five surveyed sites in the Bathurst-Orange range identified as potential tussock grassland sites (see map: http://www.environment.gov.au/biodiversity/threatened/communities/maps/pubs/152-map.pdf). However, other private and public lands need to be assessed as potential habitats for *T. mccartneyi* sp. nov., including roadside and rail verges, travelling stock routes and reserves. Assessing these areas is of importance at the moment as a new management plan for the NSW Travelling Stock Reserves was released for public comment in late 2018 [63]. Locating any remaining populations of *T. mccartneyi* sp. nov. is essential for the conservation of this species, before any further habitat losses occur.

# 5. Conclusion

Our study examines the first potential extinction of a reptile in mainland Australia and although we are not able to fully address this within the scope of our taxonomic study, we highlight the urgent need for additional research, field surveys and conservation management of GEDs in southeastern Australia. We

demonstrate the fundamental importance of rigorous scientific-based taxonomic research in conservation management. Without a stable taxonomy, including a detailed understanding of species diversity, it is impossible to develop and implement management strategies that will adequately conserve diversity. Indeed, without this knowledge, management strategies may be detrimental to a species, particularly in cases where translocation or population prioritizations are used. Such management approaches risk the introduction of competing species, hybridization and loss of cryptic species [64]. Thus, it should be a priority of conservation agencies to first resolve any taxonomic uncertainties that exist in faunal groups that are of conservation concern.

Ethics. All samples used in mtDNA and morphological sections of this study were tissue samples and voucher specimens held in museum collections. Samples for the genomics section of this study were collected with animal ethics approval granted by the University of Canberra's Committee for Ethics in Animal Experimentation (CEAE 11-14, CEAE 11-22 and CEAE 15-07). The field research and collection of these animals was undertaken under permits from the ACT Government Territory and Municipal Services (Licence to Take LT2012604; LT2012617; Licence to Import LI2011594; LI2012737) and NSW Office of Environment and Heritage (Scientific Licence Section 132c SL100756 and SL101118).

Data accessibility. Additional text and data can be found in the electronic supplementary material. mtDNA sequences are available from GENBANK, with accession numbers provided in electronic supplementary material, table S2. The mtDNA alignment, SNP data files and micro X-ray CT .PLY and landmark files are available in the Figshare digital repository and can be accessed at: https://melbourne.figshare.com/s/b2a4a1e5e87d7893ea2b.

Authors' contributions. All authors contributed to analysis and interpretation of data and drafting of the manuscript. J.M., M.H. and S.D.S. contributed to study design and concept. J.M. and M.H. undertook research on nomenclatural rules and their application. K.C. collected external morphology data, CT scans and undertook genetic and morphological analyses. DNA extraction and sequencing of historic specimen was undertaken by J.S. Phylogenomic data was collected and analysed by S.D.S., A.J.M. and B.G. The paper was written by J.M., M.H. and K.C.

Competing interests. We have no competing interests.

Funding. Research support for this project provided by the Department of Environment and Planning, ACT Government.

Acknowledgements. We thank W. Osborne, G. Waters, I. McCartney, T. McGrath, N. Clemann, C. Banks, S. Mahony, D. Gilbert, A. Lee and P. Robertson for their expertise, images and advice on GEDs. We also thank members of the Grassland Earless Dragon Recovery Team for their expert advice, support and assistance on this project. Thank you to: G. Shea and F. Tillack for their research and help tracking down the history of ZMB 740; C. Hipsley for use of VGStudio and advice on geometric morphometric analysis; J. Black for technical support with micro X-ray CT scanning at the School of Earth Sciences, University of Melbourne; B. Reynolds for discussions and information about remnant grasslands and travelling stock reserves in the Orange and Bathurst regions; and C. Clements for his assistance with access to sEXTINCT. We thank S. Mahony and J. Rowley at the Australian Museum, Sydney, and collection managers at the ANWC, Canberra, for their assistance with specimen loans and data.

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
