## [Reviewer comments · Royal Society Open Science]

Review History

RSOS-190233.R0 (Original submission)

Review form: Reviewer 1

Is the manuscript scientifically sound in its present form?

Yes

Are the interpretations and conclusions justified by the results?

Yes

Is the language acceptable?

Yes

Is it clear how to access all supporting data?

Yes

Do you have any ethical concerns with this paper?

No

Have you any concerns about statistical analyses in this paper?

I do not feel qualified to assess the statistics

Recommendation?

Accept with minor revision (please list in comments)

Comments to the Author(s)

The authors use a range of methods to conduct a taxonomic revision of an Australian agamid genus (*Tympanocryptus*). They re-describe 2 existing species, and provide descriptions for 2 new species. Importantly, they outline the conservation implications of their study, and discuss the possibility that *T. pinguicolla*, last seen in 1967, is extinct.

This is a well-executed study, and undoubtedly contains the most comprehensive set of approaches and analyses that I've ever seen for a species description. It is innovative with regard to the wide range of methods employed (mtDNA, genomics, external morphology, CT scans), and the sophistication with which the key taxonomic issues are addressed (identity of the lectotype), and resolved. It is well-written, and clearly suitable for publication in the journal.

I only have 2 relatively minor comments on the MS.

1. The title should be revised to make it clear that it is potentially the first reptile extinction on mainland Australia. This distinction is important to make, as there is one Christmas Island species listed as Extinct, and another two listed as Extinct in the Wild.

2. Given, the presumed high conservation concern for all taxa within the genus, the authors should provide the key information that will be needed for listing on the IUCN Red List (& Australian EPBC Act; which uses the IUCN criteria). This includes known information on population trends, population size, threats, estimated generation time, and geographic range (extent of occurrence, area of occupancy). It would be great if the authors could even suggest a potential Red List category for each species. This would fast-track subsequent Red Listing of these new taxa, and revising existing listings for the redefined taxa.

Review form: Reviewer 2

Is the manuscript scientifically sound in its present form?

Yes

Are the interpretations and conclusions justified by the results?

Yes

Is the language acceptable?

Yes

Is it clear how to access all supporting data?

Yes

Do you have any ethical concerns with this paper?

No

Have you any concerns about statistical analyses in this paper?

No

Recommendation?

Accept with minor revision (please list in comments)

Comments to the Author(s)

This is an excellent revision of a group of Australian lizards. It is very much state of the art in terms of the diversity and quality of the data collected and the modern treatment of those data. This is also an important example paper about how basic and thorough taxonomic research can have real world conservation implications. This is well known example in Australia and the authors have done a great job with crushing the problem. I was particularly impressed with the number of different data sets the authors used in their studies.

I think the paper is high quality in every way - the writing is clear and succinct, the figures and tables are informative and attractive, and the taxonomic detail and presentation is impressive. I have only one suggestion that the authors should consider. I did initially find reference to all the Species A-G confusing and it wasn't until I had read some bits a couple of times that I understood that Species A and B were to be designated as newly described species. That's OK, but the thing I didn't like was the reference to all the other putative species C-G. You call them "putative" species in the text sometimes but in the figures and in parts of the text you call them species. You say that they are the subject of a separate taxonomic work, but I don't think you gain anything by calling them species here. I think it would be safer to call all of these simply Clades A-G, rather than species, and then say that Clades C-G are the subject of separate taxonomic work. For these additional clades, you also did not have all the additional data, or do species delimitation analyses, which further supports in my mind that you shouldn't call them species here. Save that for the other paper. This is a very minor point and fixing it takes nothing away from this paper. Well done!

Decision letter (RSOS-190233.R0)

15-Mar-2019

Dear Dr Melville

On behalf of the Editors, I am pleased to inform you that your Manuscript RSOS-190233 entitled "Taxonomy and conservation of Grassland Earless Dragons: new species and an assessment of the first possible extinction of a reptile in Australia." has been accepted for publication in Royal Society Open Science subject to minor revision in accordance with the referee suggestions. Please find the referees' comments at the end of this email.

The reviewers and handling editors have recommended publication, but also suggest some minor revisions to your manuscript. Therefore, I invite you to respond to the comments and revise your manuscript.

• Ethics statement

- Data accessibility

If you wish to submit your supporting data or code to Dryad (<http://datadryad.org/>), or modify your current submission to dryad, please use the following link:
<http://datadryad.org/submit?journalID=RSOS&manu=RSOS-190233>

- Competing interests

- Authors' contributions

- Acknowledgements

- Funding statement

Because the schedule for publication is very tight, it is a condition of publication that you submit the revised version of your manuscript before 24-Mar-2019. Please note that the revision deadline

will expire at 00.00am on this date. If you do not think you will be able to meet this date please let me know immediately.

If your manuscript is newly submitted and subsequently accepted for publication, you will be asked to pay the article processing charge, unless you request a waiver and this is approved by

Royal Society Publishing. You can find out more about the charges at <http://rsos.royalsocietypublishing.org/page/charges>. Should you have any queries, please contact openscience@royalsociety.org.

on behalf of Professor Michael Bruford (Associate Editor) and Kevin Padian (Subject Editor)
openscience@royalsociety.org

Associate Editor Comments to Author (Professor Michael Bruford):

Thanks for submitting your manuscript to RSOS. It has been reviewed by two referees who both like it very much and recommend very minor revisions, which I concur with. Please make these revisions and your paper will be accepted. Congratulations! Mike Bruford.

Reviewer comments to Author:

Reviewer: 1

Comments to the Author(s)

The authors use a range of methods to conduct a taxonomic revision of an Australian agamid genus (*Tympanocryptus*). They re-describe 2 existing species, and provide descriptions for 2 new species. Importantly, they outline the conservation implications of their study, and discuss the possibility that *T. pinguicolla*, last seen in 1967, is extinct.

This is a well-executed study, and undoubtedly contains the most comprehensive set of approaches and analyses that I've ever seen for a species description. It is innovative with regard to the wide range of methods employed (mtDNA, genomics, external morphology, CT scans), and the sophistication with which the key taxonomic issues are addressed (identity of the lectotype), and resolved. It is well-written, and clearly suitable for publication in the journal.

I only have 2 relatively minor comments on the MS.

1. The title should be revised to make it clear that it is potentially the first reptile extinction on mainland Australia. This distinction is important to make, as there is one Christmas Island species listed as Extinct, and another two listed as Extinct in the Wild.

2. Given, the presumed high conservation concern for all taxa within the genus, the authors should provide the key information that will be needed for listing on the IUCN Red List (& Australian EPBC Act; which uses the IUCN criteria). This includes known information on population trends, population size, threats, estimated generation time, and geographic range (extent of occurrence, area of occupancy). It would be great if the authors could even suggest a potential Red List category for each species. This would fast-track subsequent Red Listing of these new taxa, and revising existing listings for the redefined taxa.

Reviewer: 2

Comments to the Author(s)

This is an excellent revision of a group of Australian lizards. It is very much state of the art in terms of the diversity and quality of the data collected and the modern treatment of those data. This is also an important example paper about how basic and thorough taxonomic research can have real world conservation implications. This is well known example in Australia and the authors have done a great job with crushing the problem. I was particularly impressed with the number of different data sets the authors used in their studies.

I think the paper is high quality in every way - the writing is clear and succinct, the figures and tables are informative and attractive, and the taxonomic detail and presentation is impressive.

I have only one suggestion that the authors should consider. I did initially find reference to all the Species A-G confusing and it wasn't until I had read some bits a couple of times that I understood that Species A and B were to be designated as newly described species. That's OK, but the thing I didn't like was the reference to all the other putative species C-G. You call them "putative" species in the text sometimes but in the figures and in parts of the text you call them species. You say that they are the subject of a separate taxonomic work, but I don't think you gain anything by calling them species here. I think it would be safer to call all of these simply Clades A-G, rather than species, and then say that Clades C-G are the subject of separate taxonomic work. For these additional clades, you also did not have all the additional data, or do species delimitation analyses, which further supports in my mind that you shouldn't call them species here. Save that for the other paper. This is a very minor point and fixing it takes nothing away from this paper. Well done!

Author's Response to Decision Letter for (RSOS-190233.R0)

See Appendix A.

Decision letter (RSOS-190233.R1)

02-Apr-2019

Dear Dr Melville,

I am pleased to inform you that your manuscript entitled "Taxonomy and conservation of Grassland Earless Dragons: new species and an assessment of first possible extinction of a reptile on mainland Australia." is now accepted for publication in Royal Society Open Science.

on behalf of Professor Michael Bruford (Associate Editor) and Kevin Padian (Subject Editor)
openscience@royalsociety.org

Appendix A

Associate Editor Comments to Author (Professor Michael Bruford):

Thanks for submitting your manuscript to RSOS. It has been reviewed by two referees who both like it very much and recommend very minor revisions, which I concur with. Please make these revisions and your paper will be accepted. Congratulations! Mike Bruford.

Reviewer comments to Author:

Reviewer: 1

Comments to the Author(s)

The authors use a range of methods to conduct a taxonomic revision of an Australian agamid genus (*Tympanocryptus*). They re-describe 2 existing species, and provide descriptions for 2 new species. Importantly, they outline the conservation implications of their study, and discuss the possibility that *T. pinguicolla*, last seen in 1967, is extinct.

In reference to the last seen in 1967 statement in the paper. It has come to our attention from one of the researchers on the recovery team for *T. pinguicolla* that there were two more high-confidence sightings for this species in 1968 and 1969, published in a local naturalist journal. Based on this we have updated the extinction analyses (dataset, methods and results section) and tightened up the wording around high- confidence sightings versus low-confidence sightings from 1988 and 1990. With these edits, including in the discussion, we make it clear that the most recent sightings are not considered of high-confidence by the recovery team and that further analyses, beyond the scope of our study, should incorporate survey effort and available habitats.

This is a well-executed study, and undoubtedly contains the most comprehensive set of approaches and analyses that I've ever seen for a species description. It is innovative with regard to the wide range of methods employed (mtDNA, genomics, external morphology, CT scans), and the sophistication with which the key taxonomic issues are addressed (identity of the lectotype), and resolved. It is well-written, and clearly suitable for publication in the journal.

I only have 2 relatively minor comments on the MS.

1. The title should be revised to make it clear that it is potentially the first reptile extinction on mainland Australia. This distinction is important to make, as there is one Christmas Island species listed as Extinct, and another two listed as Extinct in the Wild.

Title amended as requested.

2. Given, the presumed high conservation concern for all taxa within the genus, the authors should provide the key information that will be needed for listing on the IUCN

Red List (& Australian EPBC Act; which uses the IUCN criteria). This includes known information on population trends, population size, threats, estimated generation time, and geographic range (extent of occurrence, area of occupancy). It would be great if the authors could even suggest a potential Red List category for each species. This would fast-track subsequent Red Listing of these new taxa, and revising existing listings for the redefined taxa.

As requested we have added in a summary of available information for *T. lineata* and *T. osbornei* on population declines, population size and information on trends. This has been added into the remarks section of the taxonomic treatments. We have also provided references that provide more details about these factors and the threats. Although we believe this is important information for determining the conservation status of this species, we believe that it beyond the scope of a taxonomic work to make decisions or recommendations in regards to specific Red List categories for the species and that such decisions should be made in consultation with the national recovery team. Such work will be undertaken by the recovery team once the taxonomic revision has been published.

Reviewer: 2

Comments to the Author(s)

This is an excellent revision of a group of Australian lizards. It is very much state of the art in terms of the diversity and quality of the data collected and the modern treatment of those data. This is also an important example paper about how basic and thorough taxonomic research can have real world conservation implications. This is well known example in Australia and the authors have done a great job with crushing the problem. I was particularly impressed with the number of different data sets the authors used in their studies.

I think the paper is high quality in every way - the writing is clear and succinct, the figures and tables are informative and attractive, and the taxonomic detail and presentation is impressive.

I have only one suggestion that the authors should consider. I did initially find reference to all the Species A-G confusing and it wasn't until I had read some bits a couple of times that I understood that Species A and B were to be designated as newly described species. That's OK, but the thing I didn't like was the reference to all the other putative species C-G. You call them "putative" species in the text sometimes but in the figures and in parts of the text you call them species. You say that they are the subject of a separate taxonomic work, but I don't think you gain anything by calling them species here. I think it would be safer to call all of these simply Clades A-G, rather than species, and then say that Clades C-G are the subject of separate taxonomic work. For these additional clades, you also did not have all the additional data, or do

species delimitation analyses, which further supports in my mind that you shouldn't call them species here. Save that for the other paper. This is a very minor point and fixing it takes nothing away from this paper. Well done!

We see the logic of Reviewer 1's suggestion and as such we have revised the mtDNA phylogenetic section of the results. We now present just the information relevant to the Grassland Earless Dragons, have revised the distribution map and removed the species lettering from the tree. This information will be put into the second taxonomic paper on these earless dragons, which will be submitted to Royal Society Open Science in the next couple of weeks. We believe that these changes provide a better focus for the paper, taking out somewhat irrelevant details that may distract the reader. And as the reviewer suggests, these changes do not detract from the paper or lessen its importance.